# Hidden shift of the ionome of plants exposed to elevated CO$_2$ depletes minerals at the base of human nutrition

Irakli Loladze[1]*

[1]Department of Mathematics Education, The Catholic University of Daegu, Gyeongsan, Republic of Korea

**Abstract** Mineral malnutrition stemming from undiversified plant-based diets is a top global challenge. In C$_3$ plants (e.g., rice, wheat), elevated concentrations of atmospheric carbon dioxide (eCO$_2$) reduce protein and nitrogen concentrations, and can increase the total non-structural carbohydrates (TNC; mainly starch, sugars). However, contradictory findings have obscured the effect of eCO$_2$ on the ionome—the mineral and trace-element composition—of plants. Consequently, CO$_2$-induced shifts in plant quality have been ignored in the estimation of the impact of global change on humans. This study shows that eCO$_2$ reduces the overall mineral concentrations (−8%, 95% confidence interval: −9.1 to −6.9, $p<0.00001$) and increases TNC:minerals > carbon:minerals in C$_3$ plants. The meta-analysis of 7761 observations, including 2264 observations at state of the art FACE centers, covers 130 species/cultivars. The attained statistical power reveals that the shift is systemic and global. Its potential to exacerbate the prevalence of 'hidden hunger' and obesity is discussed.

## Introduction

Mankind's ultimate source of carbohydrates is atmospheric carbon dioxide (CO$_2$) converted by photosynthesis to sugars. The bulk of the terrestrial conversion of CO$_2$-to-carbohydrates is done by C$_3$ plants, which account for over three quarters of global primary production and for over 90% of Earth's plant species (*Still and Berry, 2003*). (*If not stated otherwise, hereafter, terms 'plant(s)' and 'crop(s)' refer to C$_3$ species*). When exposed to CO$_2$ concentrations twice the preindustrial level of ~280 ppm, plants increase the synthesis of carbohydrates by 19–46% (*Leakey et al., 2009*). Currently, CO$_2$ concentrations are reaching 400 ppm—the highest level since the dawn of agriculture and likely to be the highest since the rise of modern humans (*Siegenthaler et al., 2005*). Within a single human lifespan, CO$_2$ levels are projected to reach 421–936 ppm (*IPCC, 2013*). Will rising CO$_2$ concentrations—one of the most certain and pervasive aspects of global climate change—alter the quality of crops and wild plants? Will the CO$_2$-induced stimulation of carbohydrate synthesis increase the carbohydrates-to-minerals ratio in crops? Can such shifts in crop quality affect human nutrition and health?

Elevated CO$_2$ effects on plant *quantity* (productivity and total biomass) have been extensively studied and show higher agricultural yields for crops, including wheat, rice, barley, and potato. But eCO$_2$ effects on plant *quality*, and possible cascading effects on human nutrition, have been largely ignored in the estimation of the impact of eCO$_2$ on humans. Notably, *IPCC (2007, 2013)* and *AAAC Climate Science Panel (2014)* include direct CO$_2$ effects (e.g., ocean acidification) in their climate change assessments but do not mention any CO$_2$ effects on crop or wild plant quality. However, it is unwarranted to assume that plants will balance the increased carbohydrate synthesis with other adjustments to their physiology to maintain the nutritional quality for their consumers in a state of unperturbed homeostasis. The stoichiometry—the relative ratios of chemical elements—in plants is plastic and, to a considerable degree, reflects their environment (*Sterner and Elser, 2002*). However, detecting CO$_2$-induced shifts in plant quality is challenging for several reasons. First, plant quality involves

*For correspondence: loladze@asu.edu

Competing interests: The author declares that no competing interests exist.

**eLife digest** Rice and wheat provide two out every five calories that humans consume. Like other plants, crop plants convert carbon dioxide (or $CO_2$) from the air into sugars and other carbohydrates. They also take up minerals and other nutrients from the soil.

The increase in $CO_2$ in the atmosphere that has happened since the Industrial Revolution is thought to have increased the production of sugars and other carbohydrates in plants by up to 46%. $CO_2$ levels are expected to rise even further in the coming decades; and higher levels of $CO_2$ are known to lead to lower levels of proteins in plants. But less is known about the effects of $CO_2$ levels on the concentrations of minerals and other nutrients in plants.

Loladze has investigated the effect of rising $CO_2$ levels on the nutrient levels in food plants by analyzing data on 130 varieties of plants: his dataset includes the results of 7761 observations made over the last 30 years, by researchers around the world. Elevated $CO_2$ levels were found to reduce the overall concentration of 25 important minerals—including calcium, potassium, zinc, and iron—in plants by 8% on average. Furthermore, Loladze found that an increased exposure to $CO_2$ also increased the ratio of carbohydrates to minerals in these plants.

This reduction in the nutritional value of plants could have profound impacts on human health: a diet that is deficient in minerals and other nutrients can cause malnutrition, even if a person consumes enough calories. This type of malnutrition is common around the world because many people eat only a limited number of staple crops, and do not eat enough foods that are rich in minerals, such as fruits, vegetables, dairy and meats. Diets that are poor in minerals (in particular, zinc and iron) lead to reduced growth in childhood, to a reduced ability to fight off infections, and to higher rates of maternal and child deaths.

Loladze argues that these changes might contribute to the rise in obesity, as people eat increasingly starchy plant-based foods, and eat more to compensate for the lower mineral levels found in crops. Looking to the future, these findings highlight the importance of breeding food crops to be more nutritious as the world's $CO_2$ levels continue to rise.

multiple nutritional currencies, for example macronutrients (carbohydrates, protein, and fat) and micronutrients (minerals, vitamins and phytonutrients). Assessing relative changes within and among multiple currencies requires significantly more effort and funding than measuring only plant quantity (e.g., yield). Second, plant quality, including the plant ionome—all the minerals and trace-elements found in a plant (*Lahner et al., 2003*; *Salt et al., 2008*)—is inherently variable; and measurement imprecisions further amplify the variability. For example, *Stefan et al. (1997)* report the accuracy test for 39 facilities that analyzed samples of the same plant tissues: the inter-laboratory variance was 6.5% for N, but twice as large for phosphorus (P) and calcium (Ca), and reached 130% for sodium (Na). Therefore, $CO_2$-induced changes in the plant ionome (the signal) can be easily lost amid highly variable data (the noise), especially when such data are limited and sample sizes are small. However, it is important to bear in mind that a low signal-to-noise ratio *does not* imply that the signal is practically insignificant, especially if it is global and sustained—a point revisited in the 'Discussion'.

## Elusive $CO_2$ effect on the plant ionome: contradictory findings

The first empirical evidence of lower mineral content in plants exposed to $eCO_2$ appeared at least over a quarter century ago (e.g., *Porter and Grodzinski, 1984*; *Peet et al., 1986*; *O'Neill et al., 1987*). Physiological mechanisms responsible for the overall decline of plant mineral content—with expected changes being *non-uniform* across minerals—have been proposed: the increased carbohydrate production combined with other $eCO_2$ effects such as reduced transpiration (*Loladze, 2002*; *McGrath and Lobell, 2013*). However, most of the experimental evidence showing $CO_2$-induced mineral declines came from artificial facilities, mainly closed chambers and glasshouses, and many results were statistically non-significant. This led some research groups to challenge altogether the notion of lower mineral content in plants exposed to $eCO_2$ in field conditions. Such conditions are most accurately represented in Free-Air Carbon dioxide Enrichment (FACE) centers, which have been established in at least 11 countries.

In the grains of rice harvested at four FACE paddies in Japan, *Lieffering et al. (2004)* found no decline in any of the minerals but lower N content. The result disagreed with *Seneweera and Conroy (1997)*, who were the first to report lower iron (Fe) and zinc (Zn) in grains of rice grown at $eCO_2$ and warned that altered rice quality can negatively affect developing countries. *Lieffering et al. (2004)*, however, argued that the result of *Seneweera and Conroy (1997)* could be an artifact of growing rice in pots, which restrict rooting volumes. They hypothesized that in FACE studies, which provide unrestricted rooting volumes, plants would increase uptake of all minerals to balance the increased carbohydrate production. This hypothesis, however, found no support in the FACE studies of *Pang et al. (2005)* and *Yang et al. (2007)* (carried out in China and latitudinally not very far from the study in Japan), who found that $eCO_2$ significantly altered the content of several minerals in rice grains.

The contradictory results coming from these studies on rice seem perplexing, especially in light of the very robust effect that $eCO_2$ has on N in non-leguminous plants. Elevated $CO_2$ reduces N concentrations by 10–18% systemically throughout various tissues: leaves, stems, roots, tubers, reproductive and edible parts, including seeds and grains (*Cotrufo et al., 1998*; *Jablonski et al., 2002*; *Taub et al., 2008*). If the increased carbohydrate production dilutes the nutrient content in plants, why does the dichotomy seem to exist between the responses of N and minerals to $eCO_2$? In addition to the carbohydrate dilution and reduced transpiration, $eCO_2$ can further lower N concentrations in plants by: (1) reducing concentrations of Rubisco—one of the most abundant proteins on Earth that comprises a sizable N-pool in plants (*Drake et al., 1997*), and (2) inhibiting nitrate assimilation (*Bloom et al., 2010*). Hence, it is reasonable to expect the effect of $eCO_2$ on N to be larger and, thus, easier to discern than its effect on most minerals. The stronger signal for N, combined with the plentiful and less noisy data on this element, can help explain why by the end of last century the effect of $eCO_2$ on N had been already elucidated (*Cotrufo et al., 1998*), but its effect on minerals has remained elusive.

The obscure nature of the effect of $eCO_2$ on minerals becomes particularly apparent in the largest to date meta-analysis on the issue by *Duval et al. (2011)*, who fragmented data from 56 $eCO_2$ studies into 67 cases. In 47 of the cases, the effect of $eCO_2$ on minerals was statistically non-significant, that is the 95% Confidence Interval (CI) for the effect size overlapped with 0. The remaining 20 cases were statistically significant but showed no pattern: for example, Fe increased in grasses but decreased in trees, Zn increased in roots but decreased in stems, while in grains only sulfur (S) decreased. *Duval et al. (2011)* concluded: "A major finding of this synthesis is the lack of effect of $CO_2$ on crop grains nutrient concentration". This would imply laying to rest the hypothesis that $eCO_2$ consistently alters the plant ionome and would render mitigation efforts to combat declining crop mineral concentrations in the rising $CO_2$ world unnecessary. However, a closer examination of the results of *Duval et al. (2011)* reveals that every statistically significant increase in mineral concentrations was obtained by bootstrapping a sample of size 2, 4 or 5—a recipe for generating invalid 95% CIs. *Ioannidis (2005)* showed that false research findings, stemming from small sample sizes and associated low statistical power, are a persistent problem in biomedical sciences.

## 'Power failure' and the plant ionome

Calling the problem as 'power failure', *Button et al. (2013)* emphasized that the probability of a research finding to reflect a true effect drops drastically if the statistical power is reduced from 0.80 (considered as appropriate) to low levels, for example <0.30. Since the power of a statistical test drops non-linearly with the effect size, a sample size that is sufficient for detecting a 15% effect, for example a decline in N content, can be inadequate for detecting a 5% effect, for example a decline in a mineral content. Considering that the standard deviation of mineral concentrations in a plant tissue can reach 25% (*Duquesnay et al., 2000*; *Lahner et al., 2003*), the 5% effect size standardized as Cohen's $d$ is $d = 5/25 = 0.2$. A $t$ test applied for $d = 0.2$ to a sample size of 3–5—a typical size used in $eCO_2$ studies—yields the power of 0.06–0.10 (*Faul et al., 2007*). (Unfortunately, *MetaWin* (*Rosenberg et al., 2000*), a statistical package routinely used in meta-analytic and other $CO_2$ studies in ecology, provides neither a priori nor *post-hoc* power estimates.) Such a small power not only raises the probability of obtaining a false negative to 90–94% but also increases the likelihood that a statistically significant result does not reflect a true effect (*Button et al., 2013*).

## Answering questions with adequate power

As of this writing, researchers on four continents have generated data sufficient for answering with an adequate statistical power, the following questions:

1. Does $eCO_2$ shift the plant ionome? If yes, what are the direction and magnitude of shifts for individual chemical elements? How does the effect of $eCO_2$ on N compares to its effect on minerals?
2. Do FACE studies differ principally from non-FACE studies in their effect on the plant ionome?
3. Do the plant ionomes in temperate and subtropical/tropical regions differ in their response to $eCO_2$?
4. Do the ionomes of photosynthetic tissues and edible parts differ in their response to $eCO_2$? How does $eCO_2$ affect the ionomes of various plant groups (woody/herbaceous, wild/crops, $C_3/C_4$) and grains of the world's top $C_3$ cereals—wheat, rice, and barley?

## Results

For brevity, hereafter 'minerals' refer to all elements except C, hydrogen (H), oxygen (O), and N. All results are for $C_3$ plants except when noted otherwise.

### Power analysis

Plotting the effect sizes (with 95% CIs) for the 25 minerals against their respective statistical power reveals a clear pattern (*Figure 1*). In the very low power (<0.20) region, the noise completely hides the $CO_2$-induced shift of the plant ionome. In the low power region (<0.40), the shift still remains obscure. As the statistical power increases, so does the likelihood that a statistically significant result reflects true effect and, consequently, the direction and the magnitude of the $CO_2$ effect on minerals become increasingly visible in the higher power regions of the plot.

To increase the likelihood of reporting true effects, only results with the statistical power >0.40 are reported in this section. However, *Figure 1–source data 1* lists all the results together with their p-values irrespective of the statistical power (e.g., results for chromium (Cr) or the bean ionome are not shown here due to low power, but are listed in *Figure 1–source data 1*).

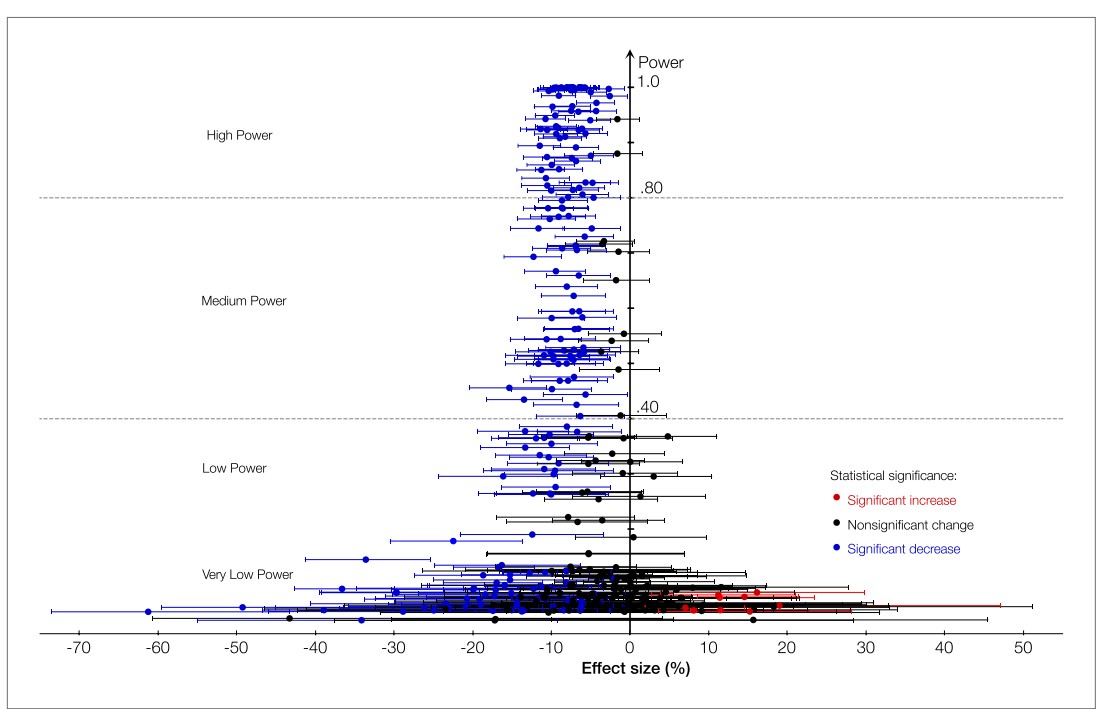

**Figure 1**. Statistical power and the effect of $CO_2$ on the plant ionome. The effect of elevated atmospheric $CO_2$ concentrations ($eCO_2$) on the mean concentration of minerals in plants plotted (with the respective 95% confidence intervals [CI]) against the power of statistical analysis. The figure reflects data on 25 minerals in edible and foliar tissues of 125 $C_3$ plant species and cultivars. The true $CO_2$ effect is hidden in the very low and the low power regions. As the statistical power increases, the true effect becomes progressively clearer: the systemic shift of the plant ionome.
The following source data are available for figure 1:

**Source data 1**. Supportive data for *Figures 1–8*.

## $CO_2$ effect on individual elements

Across all the data, $eCO_2$ reduced concentrations of P, potassium (K), Ca, S, magnesium (Mg), Fe, Zn, and copper (Cu) by 6.5–10% ($p<0.0001$) as shown on *Figure 2*. Across all the 25 minerals, the mean change was (−8%, −9.1 to −6.9, $p<0.00001$). Only manganese (Mn) showed no significant change. It is not clear whether the oxygen-evolving complex (OEC) demands for Mn separate this mineral from the pattern of declines exhibited by other minerals. Among all the measured elements, only C increased (6%, 2.6 to 10.4, $p<0.01$). The sharp difference between the responses of C and minerals to $eCO_2$ is expected if a higher carbohydrate content drives the change in the plant ionome: for most plant tissues, the dilution by carbohydrates lowers the content of minerals while having little effect on C (*Loladze, 2002*). (This also suggests that the increase in C concentrations found here could be a result of a higher content of lipids or lignin—the two sizable plant compounds that are very C-rich [~60–75% C].)

The patterns of change within edible and foliar tissues are similar: N, P, Ca, Mg, Zn, and Cu declined significantly in both tissues (*Figures 3, 4*). Aside from Mn, only K showed no significant decline in the

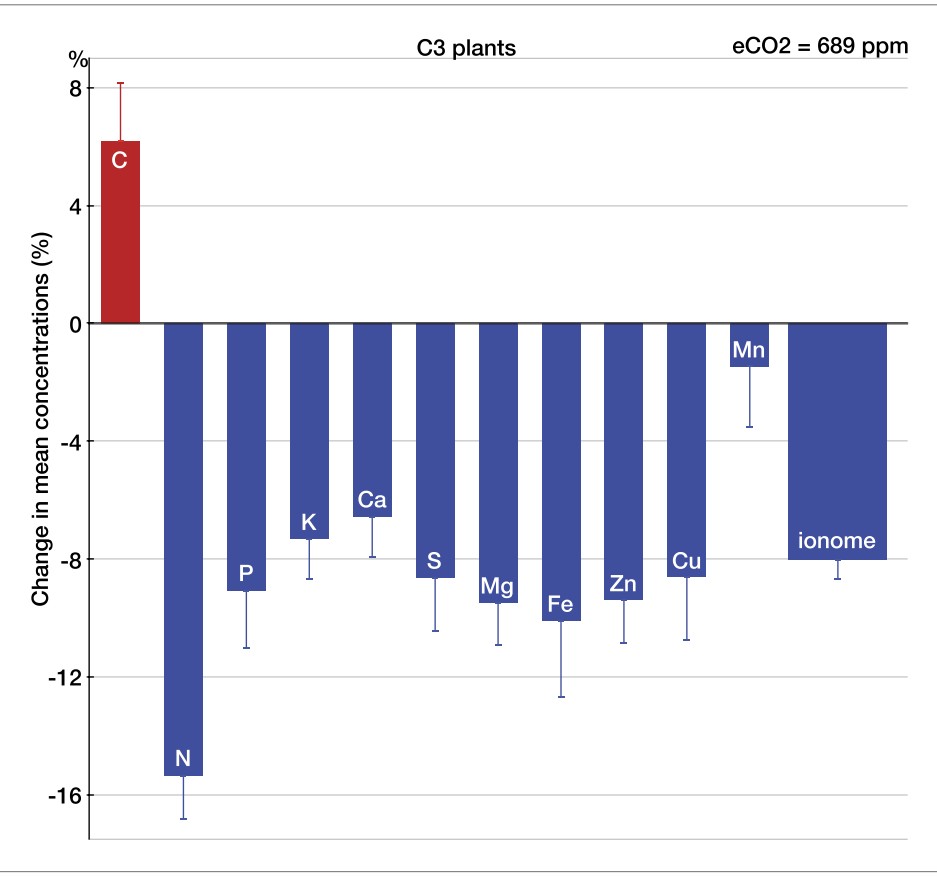

**Figure 2**. The effect of $CO_2$ on individual chemical elements in plants. Change (%) in the mean concentration of chemical elements in plants grown in $eCO_2$ relative to those grown at ambient levels. Unless noted otherwise, all results in this and subsequent figures are for $C_3$ plants. Average ambient and elevated $CO_2$ levels across all the studies are 368 ppm and 689 ppm respectively. The results reflect the plant data (foliar and edible tissues, FACE and non-FACE studies) from four continents. Error bars represent the standard error of the mean (calculated using the number of *mean* observations for each element). The number of mean and total (with all the replicates) observations for each element is as follows: C(35/169), N(140/696), P(152/836), K(128/605), Ca(139/739), S(67/373), Mg(123/650), Fe(125/639), Zn(123/702), Cu(124/612), and Mn(101/493). An element is shown individually if the statistical power for a 5% effect size for the element is >0.40. The 'ionome' bar reflects all the data on 25 minerals (all the elements in the dataset except of C and N). All the data are available at Dryad depository and at GitHub. Copies of all the original sources for the data are available upon request.

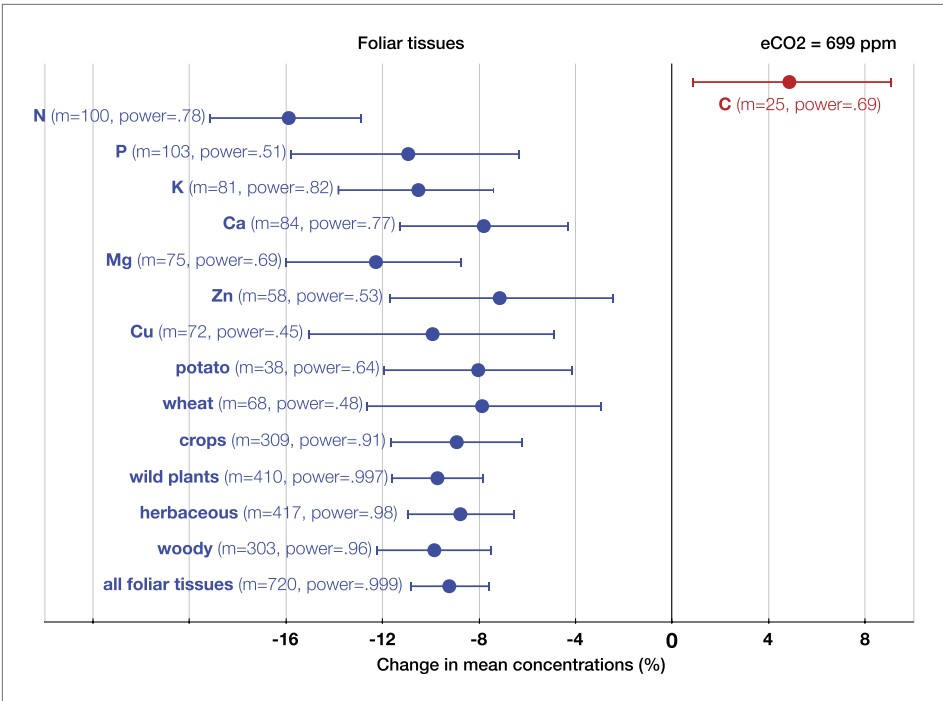

**Figure 3**. The effect of $CO_2$ on foliar tissues. Change (%) in the mean concentration of chemical elements in foliar tissues grown in $eCO_2$ relative to those grown at ambient levels. Average ambient and $eCO_2$ levels across all the foliar studies are 364 ppm and 699 ppm respectively. Error bars represent 95% CI. For each element, the number of independent mean observations, *m*, is shown with the respective statistical power. For each plant group, *m* equals the sum of mean observations over all the minerals (not including C and N) for that group. Elements and plant groups for which the statistical power is >0.40 (for a 5% effect size) are shown.

edible tissues (on *Figure 1*, it is visible as one of the only two black 95% CI in the 'High Power' region). In the foliar tissues, Mg declined the most (−12.3%, −16 to −8.7), which is congruent with the hypothesis of *McGrath and Lobell (2013)* that Mg should exhibit a larger decline in photosynthetic tissues because 'chlorophyll requires a large fraction of total plant Mg, and chlorophyll concentration is reduced by growth in elevated $CO_2$'. However, the 95% CIs for Mg and for most other minerals overlap. A richer dataset would shed more light on the issue of Mg in photosynthetic tissues.

As expected, among all elements N declined the most (−15%, −17.8 to −13.1, p<0.00001) (*Figure 2*), matching very closely previous findings (*Figures 3–6*): the 17–19% decline in leaves found by *Cotrufo et al. (1998)* and the 14% decline in seeds found by *Jablonski et al. (2002)*. Since the contents of N and protein correlate strongly in plant tissues, the lower N in edible tissues (*Figure 4*) corroborates the protein declines in crops found by *Taub et al. (2008)*.

## FACE vs non-FACE studies

With respect to the types of experiments, the $CO_2$ effect on the plant ionome is surprisingly robust: in both the FACE and the non-FACE studies $eCO_2$ significantly reduced N, P, K, Ca, S, Mg, and Zn (*Figures 5, 6*). The high cost of $CO_2$ required for running free-air experiments led to a much lower average level of $eCO_2$ in the FACE studies (560 ppm) cf. 732 ppm in the non-FACE studies. It is plausible that the lower levels of $CO_2$ in the FACE studies contributed to a smaller overall mineral decline (−6.1%, −7.8 to −4.4) cf. (−8.7%, −10.1 to −7.4) for the non-FACE studies. In both the FACE and the non-FACE studies, the overall mineral concentrations declined significantly in herbaceous plants and crops, foliar and edible tissues, including wheat and rice (*Figures 5, 6*).

## Geographical analysis

The $CO_2$ effect on the plant ionome appears to be pervasive throughout latitudes (*Figures 7, 8*). With the exception of three small centers (in Bangladesh, Japan, and the UK), the mean mineral concentrations

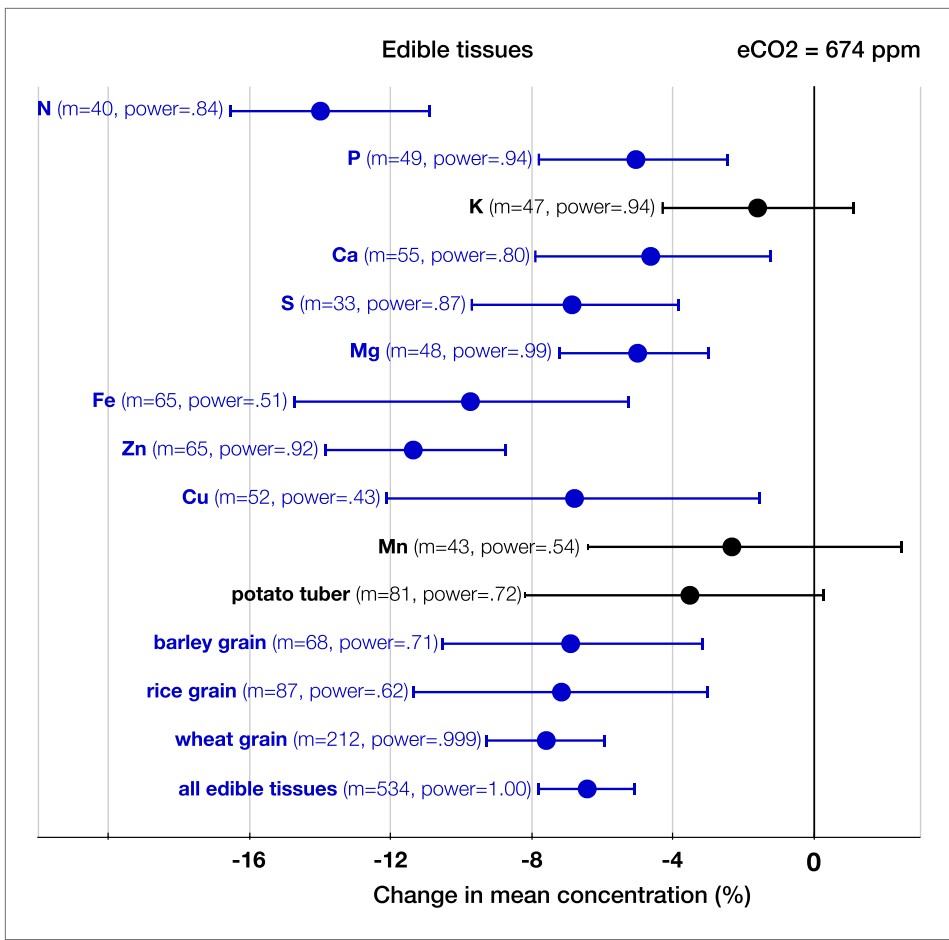

**Figure 4**. The effect of $CO_2$ on edible tissues. Change (%) in the mean concentration of chemical elements in edible parts of crops grown in $eCO_2$ relative to those grown at ambient levels. Average ambient and elevated $CO_2$ levels across all the crop edible studies are 373 ppm and 674 ppm respectively. Other details are in the legends for *Figures 2 and 3*.

declined in every FACE and open top chamber (OTC) center on four continents. The mineral decline in the tropics and subtropics (−7.2%, −10.4 to −4.0, p<0.0001) is comparable to the decline in the temperate region (−6.4%, −7.9 to −5.0, p<0.00001). A finer regional fragmentation currently is not possible due to lack of data for Africa, South America, Russia, and Canada. For many existing centers the data are limited and yield a low statistical power.

Germany leads the world in the FACE and OTC data generation with the largest number of *mean* observations of mineral concentrations (285), followed by the USA (218) (*Figure 8*). Though Australia generated only 30 mean observations, it stands out in the exceptional precision of some of its studies: the wheat experiments of *Fernando et al. (2014)* employed an unprecedented for FACE studies 48 replicates (for this reason, the study is easily identifiable on *Figure 9*).

## $CO_2$ effect on various plant groups and tissues

Since $eCO_2$ does not stimulate carbohydrate production in $C_4$ plants to a degree that it does in $C_3$ plants, one would expect a milder $CO_2$ effect on minerals for $C_4$ plants. Indeed, no statistically significant effect was found on the ionome of C4 plants (*Figure 8*). Note, however, that the very limited data on this plant group are insufficient for deducing the absence of the effect; rather, it is likely that the effect size <5% for $C_4$ plants.

The $CO_2$ effect on the $C_3$ plant ionome shows its systemic character through the analysis of various plant groups and tissues (*Figures 3, 4 and 8*). Elevated $CO_2$ reduced the overall mineral concentrations in crops (−7.2%, −8.6 to −5.6); wild (−9.7%, −11.6 to −7.8), herbaceous (−7.5%, −8.7 to −5.6), and woody (−9.6%, −12.1 to −7.6) plants; foliar (−9.2%, −10.8 to −7.6) and edible (−6.4%, −7.8 to −5.1)

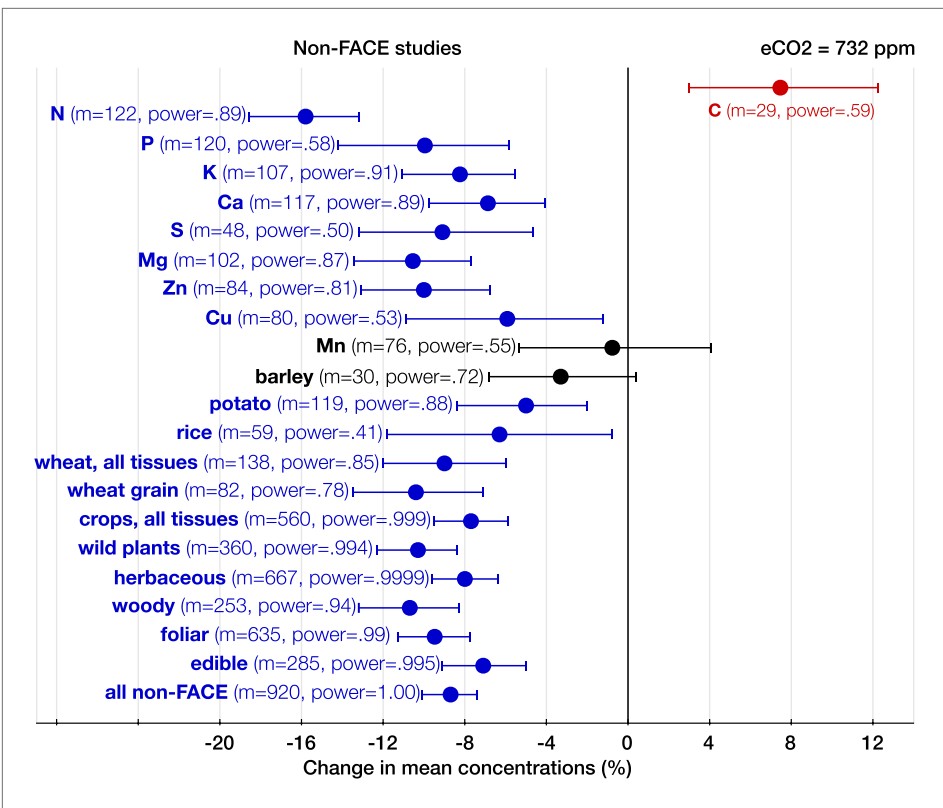

**Figure 5**. The effect of $CO_2$ in artificial enclosures. Change (%) in the mean concentration of chemical elements of plants grown in chambers, greenhouses, and other artificial enclosures under $eCO_2$ relative to those grown at ambient levels. Average ambient and $eCO_2$ levels across all the non-FACE studies are 365 ppm and 732 ppm respectively. Other details are in the legends for *Figures 2 and 3*.

tissues, including grains (−7.2%, −8.6 to −5.6). The cereal specific declines in *grains* are as follows: wheat (−7.6%, −9.3 to −5.9), rice (−7.2%, −11.3 to −3.1), and barley (−6.9%, −10.5 to −3.2) (*Figure 8*). This is notable because wheat and rice alone provide over 40% of calories to humans.

## Discussion

The analysis of all the data shows that $eCO_2$ shifts the plant ionome toward lower mineral content; the mean change across all the 25 measured minerals is (−8%, −9.1 to −6.9) (*Figure 2*). This shift, however, is hidden from low-powered statistical tests (*Figure 1*). Attaining adequate meta-analytic power reveals that the shift is:

1. Empirically robust—evident in both artificial (chambers, greenhouses) and field (FACE) conditions (*Figures 5 and 6*).
2. Geographically pervasive—found in temperate and subtropical/tropical regions (*Figures 7 and 8*).
3. Systemic—affecting herbaceous and woody plants, crops, and wild plants, photosynthetic and edible tissues, including wheat, rice, and barley grains (*Figures 3, 4 and 8*).

### Elevated $CO_2$ alters plant C:N:P:S stoichiometry

Not only does $eCO_2$ reduce the plant mineral content, but it also alters plant stoichiometry. Specifically, the effect of $eCO_2$ on N is nearly twice as large as its mean effect on minerals. The differential effect of $eCO_2$ on N (15%), and P (9%) and S (9%) translates into a ~7% reduction in the plant N:P and N:S. In contrast to the lower N and mineral content, $eCO_2$ increased C content by 6% (*Figures 2, 3 and 5*). It follows then that $eCO_2$ increases C:P and C:S by 16%, and C:N by 25% confirming the previous findings of 19–27% higher C:N in plants grown in $eCO_2$ (*Poorter et al., 1997*; *Stiling and Cornelissen, 2007*; *Robinson et al., 2012*).

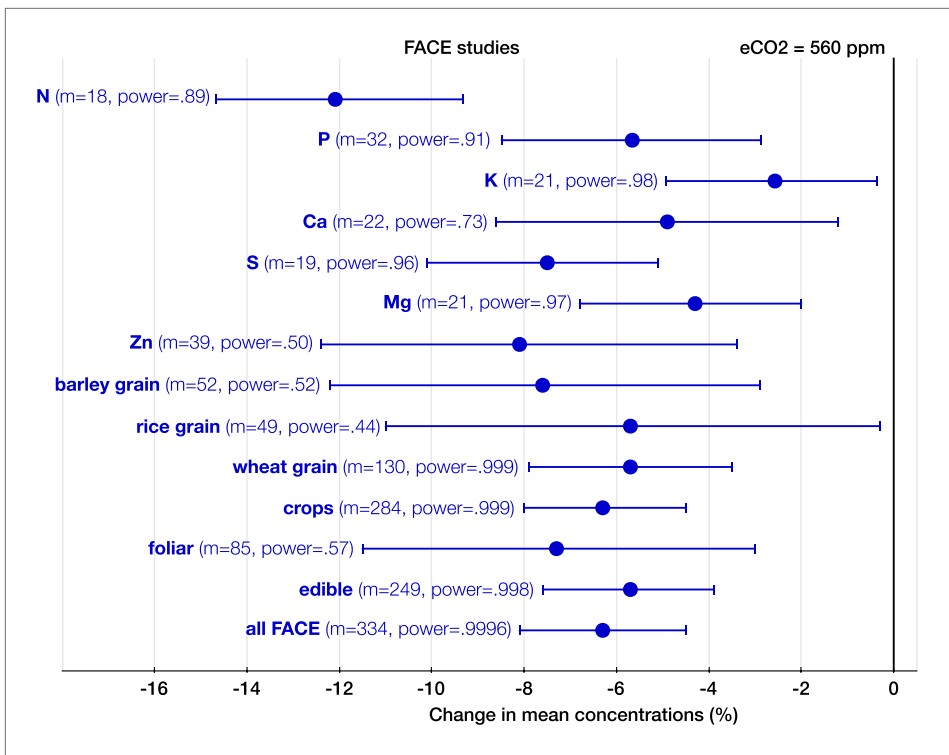

**Figure 6**. The effect of $CO_2$ at FACE centers. Change (%) in the mean concentration of chemical elements of plants grown in Free-Air Carbon dioxide Enrichments (FACE) centers relative to those grown at ambient levels. Average ambient and $eCO_2$ levels across all the FACE studies are 376 ppm and 560 ppm respectively. Other details are in the legends for **Figures 2 and 3**.

## Data scarcity

The current dataset (available at Dryad depository) suffices to show the overall shift in the plant ionome. However, it would require much richer datasets to quantify differences among the shifts of various minerals and to assess shifts in the ionomes of individual species. Unfortunately, funding hurdles for analyzing fresh and archived samples harvested at FACE centers have significantly delayed progress in this area. Only two $CO_2$ studies report selenium (Se) content (**Högy et al., 2009**, **2013**), and none report data on tin (Sn), lithium (Li), and most other trace-elements. For many of the world's popular crops, pertinent data are non-existent or very limited, including (in the descending order of calories provided to the world's population, **FAO, 2013**): maize (the top $C_4$ crop), soybeans (including oil), cassava, millet, beans, sweet potatoes, bananas, nuts, apples, yams, plantains, peas, grapes, rye, and oats.

The current data scarcity, however, should not detract our attention from what is likely to be the overarching physiological driver behind the shift in the plant ionome—the $CO_2$-induced increase in carbohydrate production and the resulting dilution by carbohydrates. Let us take a closer look at this nutritionally important issue.

## TNC:protein and TNC:minerals respond strongly to elevated $CO_2$

Carbohydrates in plants can be divided into two types: total structural carbohydrates (TSC; e.g., cellulose or fiber) that human body cannot digest, and total non-structural carbohydrates (TNC), most of which—including starch and several sugars (fructose, glucose, sucrose, and maltose)—is readily digestible and absorbed in the human gut. Hence, for humans, TNC carries the most of caloric and metabolic load of carbohydrates. Out of the two types of carbohydrates, $eCO_2$ affects stronger the latter, boosting TNC concentration by 10–45% (**Stiling and Cornelissen, 2007**; **Robinson et al., 2012**). Furthermore, $eCO_2$ tends to lower protein in plant tissues (**Taub et al., 2008**). Hence, we can reason that $eCO_2$ should exacerbate the inverse relationship found between TNC and protein

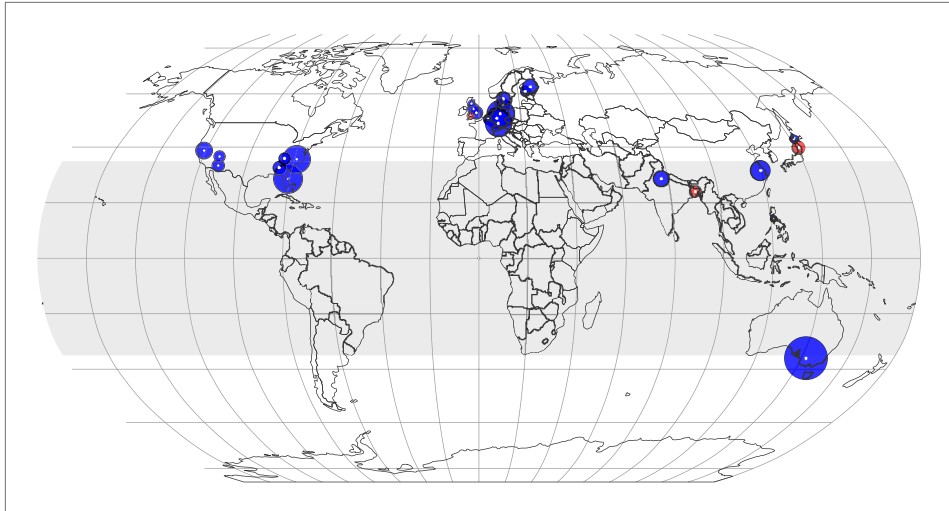

**Figure 7**. The effect of $CO_2$ at various locations and latitudes. Locations of the FACE and Open Top Chamber (OTC) centers, which report concentrations of minerals in foliar or edible tissues, are shown as white dots inside colored circles. The area of a circle is proportional to the total number of observations (counting replicates) generated by the center. If the mean change is negative (decline in mineral content), the respective circle is blue; otherwise, it is red. The figure reflects data on 21 minerals in 57 plant species and cultivars. The shaded region (between 35 N and S latitudes) represents tropics and subtropics.

(**Poorter and Villar, 1997**). Considering that TNC and protein are two out of the three primary macro-nutrients (with fats/lipids being the third), it becomes imperative to quantify changes in TNC:protein, when estimating the impact of altered plant quality on human nutrition in the rising $CO_2$ world.

Regrettably, TNC:protein is rarely reported by $CO_2$ studies; instead C:N is used as a yardstick for accessing changes in the plant quality. However, C:N poorly correlates with TNC:protein because protein is more C-rich than carbohydrates (C content in protein is 52–55% cf. 40–45% in carbohydrates). Thus, a *higher* carbohydrate:protein results in a *lower* C content. This means that $CO_2$-induced changes in nutritionally and metabolically important ratios—TNC:protein and TNC:minerals—can substantially exceed the respective changes in C:N. We can calculate changes in TNC:protein using reported changes in TNC and protein (see 'Formula for calculating percentage changes in TNC:protein and TNC:minerals' in 'Materials and methods'). *Table 1* compares $CO_2$-induced changes in C:N with respective changes in TNC:protein. It shows that $eCO_2$ can elevate TNC:protein up to fivefold higher than it does C:N.

How shifts in TNC:protein affect human nutrition is still unknown. New evidence, however, challenges "the notion that a calorie is a calorie from a metabolic perspective" by showing that changes in dietary carbohydrate:protein:fat ratios affect metabolism and weight gain in humans (**Ebbeling et al., 2012**). The new evidence supports an emerging view that while obesity is quantified as an imbalance between energy inputs and expenditures (**Hall et al., 2011**), it could also be a form of malnutrition (**Wells, 2013**), where increased carbohydrate:protein (**Simpson and Raubenheimer, 2005**) and excessive carbohydrate consumption (**Taubes, 2013**) could be possible culprits.

## Absolute $CO_2$ effect on TNC. Spoonful of sugars for everyone?

The baseline TNC content in plant tissues varies widely. In grains and tubers, it is very high, 50–85% of dry mass (DM). Therefore, in these tissues a percentage increase in TNC is arithmetically limited (e.g., a 60% increase is impossible). However, even a modest percentage increase in TNC-rich tissues can be nutritionally meaningful in absolute terms. For example, the FACE study of **Porteaus et al. (2009)** reports a 7–8% increase in starch concentrations in wheat grains, which translates to ~4 g of additional starch per 100 g DM. In contrast to grains and tubers, the baseline TNC level in photosynthetic tissues is small (usually <25%), which makes large TNC increases possible. For example, **Teng et al. (2006)** reports that $eCO_2$ increased TNC by 76% in leaves of *Arabidopsis thaliana*. What is interesting here is that in *absolute* terms (per 100 g DM) the ~5 g TNC increase in *Arabidopsis thaliana* is comparable to the ~4 g TNC increase in wheat grains.

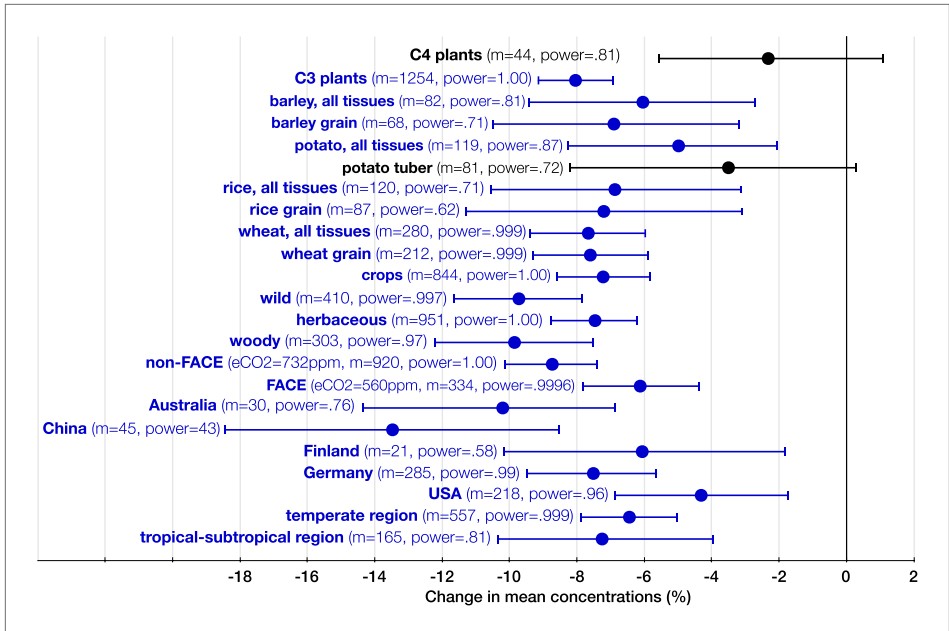

**Figure 8**. The systemic aspect of the $CO_2$ effect. Change (%) in the mean concentration of minerals in plants grown in $eCO_2$ relative to those grown at ambient levels. All the results in the figure reflect the combined data for the foliar and the edible tissues. The number of total *mean* observations (*m*) for all the measured minerals across all the studies for each crop/plant group, experiment type, country, or region is shown with the respective statistical power. Country specific and regional results reflect all the FACE and Open Top Chamber (OTC) studies carried in any given country/region. The number of total observations (with replicates) for all the minerals (not counting C and N) for each country is as follows: Australia (926), China (193), Finland (144), Germany (908), and USA (1156). Other details are in the legends for **Figures 2 and 3**.

More generally, $CO_2$ studies show that—irrespective of the baseline TNC content—$eCO_2$ tends to boost TNC by a few grams (1–8 g) per 100 g DM of plant tissue (*Poorter et al., 1997*; *Keutgen and Chen, 2001*; *Katny et al., 2005*; *Erbs et al., 2010*; *Azam et al., 2013*). Note that such an infusion of carbohydrates into plant tissues, all else being equal, dilutes the content of other nutrients by ~1–7.4%. Let us compare the dilution with its pragmatic and easily graspable analog—adding a spoonful of sugar-and-starch mixture. **Table 2** shows that the $CO_2$ effect on TNC:protein and TNC:minerals is stoichiometrically similar to the effect of adding a spoonful of carbohydrates to every 100 g DM of plant tissue.

Clearly, adding a spoonful of sugar sporadically to one's diet is not a cause for concern. However, the inescapable pervasiveness of globally rising atmospheric $CO_2$ concentrations raises new questions: What are health consequences, if any, of diluting every 100 g DM of raw plant products with a spoonful of starch-and-sugar mixture? What are the consequences if the dilution is not sporadic but unavoidable and lifelong? These questions are better left for nutritionists, but it is worth noting that **WHO (2014)** conditionally recommends that intake of free sugars not exceed 5% of total energy, which is equivalent to 5–8 teaspoons of sugar for a typical 2000–3000 kcal/day diet.

Below, I shift focus on a direct consequence of the $CO_2$-induced increase in carbohydrate production—the mineral decline in plant tissues, and explore its potential effect on human nutrition.

## Plant minerals and 'hidden hunger'

'Hidden hunger'—stems from poorly diversified plant-based diets meeting caloric but not nutritional needs. It is currently the world's most widespread nutritional disorder (**Kennedy et al., 2003**; **Welch and Graham, 2005**). It lowers the GDP of the most afflicted countries by 2–5% and is partly responsible for their Third World status (**WHO, 2002**; **Stein, 2009**). A paradoxical aspect of 'hidden hunger' is that the minuscule amount of minerals, which a human body requires, could be provided easily and inexpensively—at least in theory—to all people in need by fortifying foods with minerals. However, in practice, such required mineral levels do not reach large parts of the world's community.

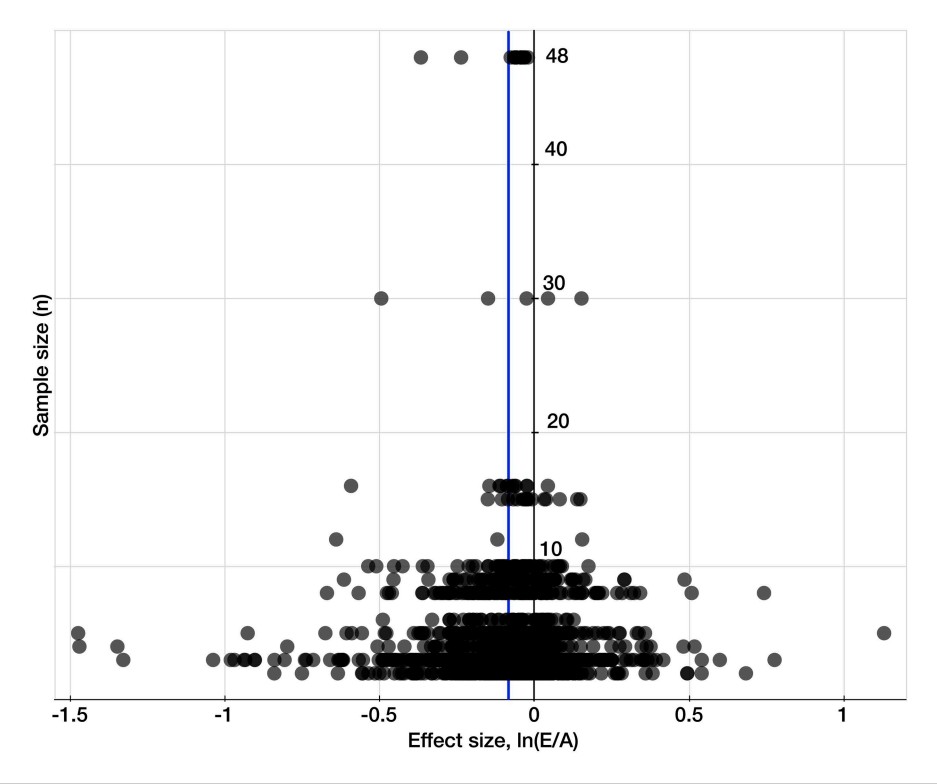

**Figure 9**. Testing for publication bias. A funnel plot of the effect size (the natural log of the response ratio) plotted against the number of replicates/sample sizes ($n$) for each study and each mineral in the dataset for $C_3$ plants. The plot provides a simple visual evaluation of the distribution of effect sizes. The blue line represents the mean effect size of $eCO_2$ on mineral concentrations: the decline of 8.39% (yielding the decline of 8.04% when back transferred from the log-form). The symmetrical funnel shape of the plot around the mean effect size indicates the publication bias in the dataset is insignificant (**Egger et al., 1997**).

The case of iodine is illustrative: although iodized table salt nearly wiped out iodine deficiency in the industrialized world, a billion people still have no regular access to it, making iodine deficiency the leading cause of preventable brain damage, cretinism, and lower IQ in children (**Welch and Graham, 1999**; **WHO, 2002**). Hence, the reality of logistic, economic, and cultural hurdles for fortification leaves the natural and bioavailable mineral content in food, and in plants in particular, to be the major, and sometimes the only, consistent mineral supply for a large part of mankind (**White and Broadley, 2009**; **Bouis and Welch, 2010**). This supply, unfortunately, is suboptimal for human nutrition with some of the consequences outlined below.

Every third person in the world is at risk of inadequate Zn intake with its deficiency substantially contributing to stunting, compromised immunity, and child mortality (**Brown et al., 2001**; **UNICEF, 2009**). Iron deficiency affects at least 2 billion people and is the leading cause of anemia that increases maternal mortality (**WHO, 2002**; **UNICEF, 2009**). Millions are Ca, Mg, and Se deficient (**Stein, 2009**; **White and Broadley, 2009**), including some population segments of developed countries (**Rayman, 2007**; **Khokhar et al., 2012**). Ironically, a person can be obese *and* mineral undernourished—the so called 'hunger-obesity paradox' (**Scheier, 2005**), for example the many homeless in the US who rely on "cheap and energy-dense but low-nutrient" foods (**Koh et al., 2012**). With every third adult in the world being overweight or obese (**Keats and Wiggins, 2014**), WHO ranks both mineral undernutrition and obesity among the top 20 global health risks (**WHO, 2002**; **Hill et al., 2003**; **Stein, 2009**). While the role of mineral deficiency in obesity is still unclear, intriguing links have been found between the lower blood serum concentrations of Ca, Cr, Fe, Mg, Mn, Se, Zn, and increased body mass index (BMI), with most of the findings appearing in the last decade (**Singh et al., 1998**; **Martin et al., 2006**; **Arnaud et al., 2007**; **García et al., 2009**; **Payahoo et al., 2013**; **Yerlikaya et al., 2013**).

**Table 1.** Comparing the effects of $CO_2$ on two plant quality indicators.

| Study/species | C:N (%) | TNC:protein (%) | Reference |
|---|---|---|---|
| *Arabidopsis thaliana* | 25 | 125 | *Teng et al. (2006)* |
| *Bromus erectus* | 6 | 26 | *Roumet et al. (1999)\** |
| *Dactylis glomerata* | 17 | 53 | *Roumet et al. (1999)\** |
| wheat grain (low N) | −10 | 47 | *Porteaus et al. (2009)* |
| wheat grain (high N) | −18 | 7 | *Porteaus et al. (2009)* |
| wheat grain | 9 | 6 | *Högy et al. (2009)* |
| 27 $C_3$ species | 28 | 90 | *Poorter et al. (1997)* |
| meta-analysis | 25 | 54 | *Robinson et al. (2012)* |
| meta-analysis | 27 | 39 | *Stiling and Cornelissen (2007)* |

$CO_2$-induced changes (%) in C:N (a quality indicator often used in $CO_2$ studies) and in TNC:protein (a rarely used but nutritionally important indicator) for wheat grains and for foliar tissues of various plants. The results shows that in the same plant tissue, $eCO_2$ can increase TNC:protein up to several-fold > C:N. Significant $CO_2$-induced shifts in the ratio of major macronutrients are probable. Hence, it is important for $CO_2$ studies to start accessing and reporting changes in TNC:protein.
\*in lieu of protein, N content is used.

How can the $CO_2$-induced depletion of minerals in crops affect humans? I emphasize that the impact of $CO_2$-induced shifts in the quality of crops on human health is far from settled. The purpose of what follows is not to make definitive claims but to stimulate research into this important but unresolved issue.

**Table 2.** Comparing the effect of $CO_2$ to the effect of adding 'a spoonful of sugars.'

| Plant quality indicator | Effect of adding 5g of TNC (%) | Effect of elevated $CO_2$ (%) |
|---|---|---|
| **Grains and tubers:** | | |
| TNC | 2.6 | 1 to 15 |
| TNC:protein | 7 | 6 to 47 |
| TNC:minerals | 7 | 6 to 28 |
| protein | −4.8 | −14 to −9 |
| minerals | −4.8 | −10 to −5 |
| **Foliar tissues:** | | |
| TNC | 27 | 15 to 75 |
| TNC:protein | 33 | 26 to 125 |
| TNC:minerals | 33 | 24 to 98 |
| protein | −4.8 | −19 to −14 |
| minerals | −4.8 | −12 to −5 |

Changes (%) in various plant quality indicators caused by: (1) Adding a teaspoon of TNC (~5g of starch-and-sugars mixture) per 100g of dry mass (DM) of plant tissue, an:d (2) growing plants in twice-ambient $CO_2$ atmosphere. Changes due to the addition of TNC are calculated assuming:the baseline TNC content of 65% for grains and tubers, and 15% for foliar tissues. The C content is assumed to be ~42% for plant tissues and TNC.

## Stoichiometric thought experiment

A randomized controlled trial for a human diet based exclusively (directly or indirectly) on plants grown in $eCO_2$ is unlikely and ethically questionable; and even if feasible, the trial might take years to generate results. In lieu of relevant data, we can employ a thought experiment. While such 'experiments' are usually reserved for physical sciences, any living system, notwithstanding its complexity, adheres to simple but irrefutable elemental mass balance, which can help us to elucidate plausible scenarios.

For simplicity, let us focus on one question: how can a 5% reduction in the plant mineral content affect human nutrition? Thus, we ignore other potential or likely $CO_2$ effects: for example higher agricultural yields; altered concentrations of lipids, vitamins, and polyphenols; substantially higher TNC:protein and TNC:minerals; differential $C_3$ and $C_4$ plant responses; changes in the phytate content that affects mineral bioavailability (*Manoj-Kumar, 2011*); and multiplicative health effects of the concomitant declines of many minerals in the same tissue.

Suppose that starting tomorrow and without our knowledge, the baseline mineral content of all plants on Earth drops by 5%. A self-evident but easily overlooked mass-balance law tells us that neither thermal nor mechanical processing of

raw plants enriches them with minerals (i.e., transmutations are impossible). Thus, the mineral decline in raw crops will follow into plant-based foods (except for a few food items that are fortified with certain minerals in some countries).

We can safely assume that the individuals, whose dietary intake of each essential mineral has exceeded the recommended dietary intake (RDI) by >5%, will be unaffected by the depletion. This leaves us with the majority of the human population, whose diet is either at risk of deficiency or already deficient in atleast one mineral (*WHO, 2002*; *Kennedy et al., 2003*; *Stein, 2009*). Though a human body can synthesize complex compounds (e.g., vitamins K and D, non-essential amino acids), the mass balance low implies that *no organism can synthesize any amount of any mineral*. Therefore, to compensate for the mineral deficit, an organism has to increase mineral intake (or, otherwise, endure the consequences of the deficit). Taking supplements or intentionally shifting one's diet toward mineral-rich foods, for example animal products, can eliminate the deficit. Such dietary changes, however, presuppose behavioral adjustments on the part of the individuals who are aware of their mineral deficiency and have both the means and motivation to address it. A simpler way to compensate for the mineral deficit is to *increase food intake*, whether consciously or not. (The notion of compensatory feeding is not entirely alien—herbivores *do* increase consumption by 14–16%, when consuming plants grown in $eCO_2$; *Stiling and Cornelissen, 2007*; *Robinson et al., 2012*).

For a calorie deficient person, eating 5% more (to be exact 5.26%, because 1.0526*.95 ≈ 1) is likely to be beneficial. However, for a calorie sufficient but mineral deficient person, eating 5% more could be detrimental. The dynamic mathematical model of human metabolism, which links weight changes to dietary and behavioral changes (*Hall et al., 2011*), can help to quantify the effect of a prolonged 5% increase in food intake. When parameterized with anthropometric data for an average moderately active American female (age 38, height 163 cm, weight 76 kg, BMI 28.6, energy intake 2431 kcal/day [10171 kJ]) (*Fryar et al., 2012*; *CIA, 2013*), the model outputs a weight gain of 4.8 kg over a 3-year period, provided all other aspects of behavior and diet remain unchanged. For a male, the respective weight gain is 5.8 kg. The results are congruent with *Hill et al. (2003)*, who argued that a 4–5% difference in total daily energy intake, a mere 100 kcal/day, could be responsible for most weight gain in the population.

The above 'experiment' suggests that a systemic and sustained 5% mineral depletion in plants can be nutritionally significant. While the rise in the atmospheric $CO_2$ concentration is expected to be nearly uniform around the globe, its impact on crop quality might unequally affect the human population: from no detrimental effects for the well-nourished to potential weight gain for the calorie-sufficient but mineral-undernourished.

## Has rising $CO_2$ already altered the plant ionome?

The rise in $CO_2$ levels over the last 18–30 years has already been implicated in the two effects that can influence the plant ionome: higher C assimilation and plant growth (*Donohue et al., 2013*), and lower transpiration (*Keenan et al., 2013*). Considering that over the last 250 years, the atmospheric $CO_2$ concentration has increased by 120 ppm—an increase that is not far from the mean 184 ppm enrichment in the FACE studies—it is plausible that plant quality has changed. Indeed, declines in mineral concentrations have been found in wild plants and in crop fruits, vegetables, and grains over 22–250 years (*Penuelas and Matamala, 1993*; *Duquesnay et al., 2000*; *Davis et al., 2004*; *Ekholm et al., 2007*; *Fan et al., 2008*; *Jonard et al., 2009*). While the mineral declines in crops can be an unintended consequence of the Green Revolution that produced high-yield cultivars with altered mineral content (*Davis et al., 2004*; *Fan et al., 2008*), the reason for the mineral declines in wild plants cannot be attributed to it.

Can $eCO_2$ directly affect human health? *Hersoug et al. (2012)* proposed that rising $CO_2$ promotes weight gains and obesity in the human population directly (via breathing) by reducing the pH of blood and, consequently, increasing appetite and energy intake. Weight gain has been observed in wild mammals, lab animals, and humans over the last several decades (*Klimentidis et al., 2011*). However, it is not clear what role, if any, the rising $CO_2$ could have played either directly (breathing) or indirectly (altered plant quality). And disentangling the rising $CO_2$ effect from other plausible factors currently does not seem feasible due to scarce data. This brings us to the broader issue of detecting—amid high local noise—signals that are small in their magnitude but global in their scope.

## Hidden shifts of global change

While some scientific areas (e.g., genomics, bioinformatics) have experienced a data deluge, many areas of global change, including the issue of shifting plant quality, have been hindered by chronic

data scarcity. Fortunately, researchers worldwide have been steadily generating data on the effects of $eCO_2$ on the chemical composition of plants. It is their collective efforts that have made it possible to reveal the $CO_2$-induced shift in the plant ionome.

Human activities profoundly alter the biogeochemical cycle not only of C but also of N, P, and S, which are central to all known life forms. It is plausible that other subtle global shifts in the physiology and functioning of organisms lurk amid highly noisy data. The small magnitude of such shifts makes them hard to detect and easy to dismiss. But by virtue of being global and sustained, the shifts can be biologically potent. Revealing hidden shifts requires plentiful data to attain sufficient statistical power. (For example, *Rohde et al. (2013)* analyzed 14 million *mean* monthly local temperature records to uncover the 1.5°C rise in the global average temperature since 1753—undoubtedly a potent but a very small change relative to the variations of tens of degrees in local temperature.)

New data on the effects of $eCO_2$ on plant quality (e.g., minerals, TNC: protein, TNC:minerals, lipids, bioavailability of nutrients) can be generated very cost-efficiently by analyzing fresh and archived plant samples collected at FACE centers worldwide (the project leaders of many centers are keen to share such samples; PS Curtis, BA Kimball, R Oren, PB Reich, C Stokes; IL personal communication, July, 2006). With regard to minerals, the application of the high-throughput techniques of ionomics (*Salt et al., 2008*) can generate rich phenotypic data that can be linked with functional genomics. Such analyses will shed more light on changes in plant quality in the rising $CO_2$ world. Anticipating and assessing such changes will help not only in mitigating their effects but also in steering efforts to breed nutritionally richer crops for the improvement of human health worldwide.

## Materials and methods

### Search for data

I searched Google Scholar, Google, PubMed, the ISI Web of Science, AGRICOLA, and Scopus to find relevant articles with sensible combinations of two or more of the following search-words: elevated, rising, $CO_2$, carbon dioxide, ppm, FACE, effects, content, concentration, %, mg, dry matter, micronutrients, plant(s), crop(s), tree(s), $C_3$, $C_4$, foliar, leaves, grains, seeds, tubers, fruits, minerals, chemical elements, and names/symbols of various chemical elements (e.g., zinc/Zn). I found additional studies from references in the articles identified in the initial searches.

### Study suitability and data selection criteria

Among all plant tissues for which mineral concentrations are reported in the literature, the most abundant data are on foliar tissues (leaves, needles, shoots), and—for herbaceous plants—on above ground parts. Hence, focusing on the foliar tissues and above ground parts allows one to maximize the number of *independent* observations of the effect of $eCO_2$ on each mineral. Although the data on edible parts of crops are scarcer, a dataset on crop edible tissues was compiled due to their direct relevance for human nutrition.

The following objective and uniform criteria were applied for deciding which studies to include into the dataset: (1) a study grew plants at two or more $CO_2$ levels, (2) a study directly measured the content of one or more minerals in foliar or edible plant tissues at low (ambient) and high (elevated) $CO_2$ levels, and (3) a study reported either absolute concentrations at each treatment or relative change/ lack thereof in the concentrations for each mineral between treatments. Studies that indirectly deduced mineral concentrations, reported data on N but not on any mineral, exposed only a part (e.g., a branch) of the plant, used super-elevated or uncontrolled $CO_2$ levels were not included. *Table 3* lists all the studies together with their respective species/cultivars and $CO_2$ enrichment levels (the dataset with all the details is deposited at Dryad and GitHub). When a study reported the low $CO_2$ level as 'ambient' with no specific numerical values, then I used the Keeling curve to approximate the ambient $CO_2$ level for the year the study was carried out.

The following data-inclusion rules were applied to the studies with multiple co-dependent datasets for the foliar dataset: (1) the lowest and the highest $CO_2$ levels for studies with multiple $CO_2$ levels, (2) the control and single-factor $CO_2$ for studies with environmental co-factors (e.g., observations from combined $eCO_2$ and ozone experiments were excluded), (3) the highest nutrient regime when the control could not be identified in a study with multiple nutrient co-factors, (4) the last point, that is the longest exposure to ambient/$eCO_2$ for studies with time series, (5) the most mature needles/leaves for studies reporting foliar tissues of various ages. If, in rare instances, a publication reported three

**Table 3.** Studies covered in the meta-analysis of $CO_2$ effects on the plant ionome.

| Species | Common name | Crop | +CO2 | Country | Reference |
|---------|-------------|------|------|---------|-----------|
| *Acer pseudoplatanus* | maple tree | No | 260 | | *Overdieck, 1993* |
| *Acer rubrum* | red maple tree | No | 200 | USA | *Finzi et al., 2001* |
| *Agrostis capillaris* | grass | No | 340 | UK | *Baxter et al., 1994* |
| *Agrostis capillaris* | grass | No | 250 | | *Newbery et al., 1995* |
| *Alnus glutinosa* | alder tree | No | 350 | UK | *Temperton et al., 2003* |
| *Alphitonia petriei* | rainforest tree | No | 440 | | *Kanowski, 2001* |
| *Ambrosia dumosa* | shrub | No | 180 | USA | *Housman et al., 2012* |
| *Arabidopsis thaliana* | thale cress | No | 450 | | *Niu et al., 2013* |
| *Arabidopsis thaliana* | thale cress | No | 330 | | *Teng et al., 2006* |
| *Betula pendula* 'Roth' | birch tree | No | 349 | Finland | *Oksanen et al., 2005* |
| *Bouteloua curtipendula* | grass | No | 230 | | *Polley et al., 2011* |
| *Bromus tectorum* | cheatgrass | No | 150 | | *Blank et al., 2006* |
| *Bromus tectorum* | cheatgrass | No | 150 | | *Blank et al., 2011* |
| *Calluna vulgaris* | heather shrub | No | 200 | | *Woodin et al., 1992* |
| *Cercis canadensis* | red bud tree | No | 200 | USA | *Finzi et al., 2001* |
| *Chrysanthemum morifolium* | chrysanth | No | 325 | | *Kuehny et al., 1991* |
| *Cornus florida* | dogwood tree | No | 200 | USA | *Finzi et al., 2001* |
| *Fagus sylvatica* | beech tree | No | 260 | | *Overdieck, 1993* |
| *Fagus sylvatica* | beech tree | No | 300 | | *Rodenkirchen et al., 2009* |
| *Festuca pratensis* | meadow fescue | No | 320 | | *Overdieck, 1993* |
| *Festuca vivipara* | grass | No | 340 | UK | *Baxter et al., 1994* |
| *Flindersia brayleyana* | rainforest tree | No | 440 | | *Kanowski, 2001* |
| *Galactia elliottii* | Elliott's milkpea | No | 325 | USA | *Hungate et al., 2004* |
| *Larix kaempferi* | larch tree | No | 335 | Japan | *Shinano et al., 2007* |
| *Lepidium latifolium* | peppergrass | No | 339 | | *Blank and Derner, 2004* |
| *Liquidambar styraciflua* | sweetgum tree | No | 200 | USA | *Finzi et al., 2001* |
| *Liquidambar styraciflua* | sweetgum tree | No | 167 | USA | *Johnson et al., 2004* |
| *Liquidambar styraciflua* | sweetgum tree | No | 156–200 | USA | *Natali et al., 2009* |
| *Liriodendron tulipifera* | tulip tree | No | 325 | | *O'Neill et al., 1987* |
| *Lolium perenne* | grass | No | 320 | | *Overdieck, 1993* |
| *Lolium perenne* | grass | No | 290 | Germany | *Schenk et al., 1997* |
| *Lupinus albus* | white lupin | No | 550 | | *Campbell and Sage, 2002* |
| *Lycium pallidum* | shrub | No | 180 | USA | *Housman et al., 2012* |
| *Nephrolepis exaltata* | fern | No | 650 | | *Nowak et al., 2002* |
| *Pelargonium x hortorum* 'Maverick White' | geranium | No | 330 | | *Mishra et al., 2011* |
| *Picea abies* 'Karst.' | spruce tree | No | 350 | | *Pfirrmann et al., 1996* |
| *Picea abies* 'Karst.' | spruce tree | No | 300 | | *Rodenkirchen et al., 2009* |
| *Picea abies* 'Karst.' | spruce tree | No | 300 | | *Weigt et al., 2011* |
| *Picea rubens* | spruce tree | No | 350 | | *Shipley et al., 1992* |
| *Pinus ponderosa* | pine tree | No | 346 | USA | *Walker et al., 2000* |
| *Pinus ponderosa* 'Laws.' | pine tree | No | 350 | USA | *Johnson et al., 1997* |
| *Pinus sylvestris* | pine tree | No | 331 | | *Luomala et al., 2005* |

*Table 3. Continued on next page*

Table 3. Continued

| Species | Common name | Crop | +CO2 | Country | Reference |
|---|---|---|---|---|---|
| Pinus sylvestris | pine tree | No | 225 | Finland | Utriainen et al., 2000 |
| Pinus taeda | loblolly pine tree | No | 200 | USA | Finzi et al., 2001 |
| Pinus taeda | pine tree | No | 200 | USA | Natali et al., 2009 |
| Poa alpina | grass | No | 340 | UK | Baxter et al., 1994 |
| Poa alpina | grass | No | 340 | UK | Baxter et al., 1997 |
| Pteridium aquilinum | fern | No | 320 | | Zheng et al., 2008 |
| Pteridium revolutum | fern | No | 320 | | Zheng et al., 2008 |
| Pteris vittata | fern | No | 320 | | Zheng et al., 2008 |
| Quercus chapmanii | oak tree | No | 350 | USA | Natali et al., 2009 |
| Quercus geminata | oak tree | No | 350 | USA | Johnson et al., 2003 |
| Quercus geminata | oak tree | No | 350 | USA | Natali et al., 2009 |
| Quercus myrtifolia | oak tree | No | 350 | USA | Johnson et al., 2003 |
| Quercus myrtifolia | oak tree | No | 350 | USA | Natali et al., 2009 |
| Quercus suber | cork oak tree | No | 350 | | Niinemets et al., 1999 |
| Schizachyrium scoparium | grass | No | 230 | | Polley et al., 2011 |
| Sorghastrum nutans | grass | No | 230 | | Polley et al., 2011 |
| Sporobolus kentrophyllus | grass | No | 330 | | Wilsey et al., 1994 |
| Trifolium alexandrinum 'Pusa Jayant' | berseem clover | No | 250 | India | Pal et al., 2004 |
| Trifolium pratense | red clover | No | 320 | | Overdieck, 1993 |
| Trifolium repens | white clover | No | 320 | | Overdieck, 1993 |
| Trifolium repens | white clover | No | 290 | Germany | Schenk et al., 1997 |
| Trifolium repens | white clover | No | 615 | | Tian et al., 2014 |
| Trifolium repens 'Regal' | white clover | No | 330 | | Heagle et al., 1993 |
| Vallisneria spinulosa | macrophyte | No | 610 | | Yan et al., 2006 |
| Apium graveolens | celery | Yes | 670 | | Tremblay et al., 1988 |
| Brassica juncea 'Czern' | mustard | Yes | 500 | India | Singh et al., 2013 |
| Brassica napus 'Qinyou 8' | rapeseed | Yes | 615 | | Tian et al., 2014 |
| Brassica napus 'Rongyou 10' | rapeseed | Yes | 615 | | Tian et al., 2014 |
| Brassica napus 'Zhongyouza 12' | rapeseed | Yes | 615 | | Tian et al., 2014 |
| Brassica napus 'Campino' | oilseed rape | Yes | 106 | Germany | Högy et al., 2010 |
| Brassica rapa 'Grabe' | turnip | Yes | 600 | | Azam et al., 2013 |
| Citrus aurantium | orange tree | Yes | 300 | USA | Penuelas et al., 1997 |
| Citrus madurensis | citrus tree | Yes | 600 | | Keutgen and Chen, 2001 |
| Cucumis sativus | cucumber | Yes | 650 | | Peet et al., 1986 |
| Daucus carota 'T-1-111' | carrot | Yes | 600 | | Azam et al., 2013 |
| Fragaria x ananassa | strawberry | Yes | 600 | | Keutgen et al., 1997 |
| Glycine max 'Merr.' | soybean | Yes | 360 | USA | Prior et al., 2008 |
| Glycine max 'Merr.' | soybean | Yes | 200 | | Rodriguez et al., 2011 |
| Gossypium hirsutum 'Deltapine 77' | cotton | Yes | 180 | USA | Huluka et al., 1994 |
| Hordeum vulgare | barley | Yes | 175 | Germany | Erbs et al., 2010 |
| Hordeum vulgare 'Alexis' | barley | Yes | 334 | Germany | Manderscheid et al., 1995 |

Table 3. Continued on next page

Table 3. Continued

| Species | Common name | Crop | +CO2 | Country | Reference |
|---|---|---|---|---|---|
| *Hordeum vulgare* 'Arena' | barley | Yes | 334 | Germany | *Manderscheid et al., 1995* |
| *Hordeum vulgare* 'Europa' | barley | Yes | 400 | | *Haase et al., 2008* |
| *Hordeum vulgare* 'Iranis' | barley | Yes | 350 | | *Pérez-López et al., 2014* |
| *Hordeum vulgare* 'Theresa' | barley | Yes | 170 | Germany | *Wroblewitz et al., 2013* |
| *Lactuca sativa* 'BRM' | lettuce | Yes | 308 | | *Baslam et al., 2012* |
| *Lactuca sativa* 'Mantilla' | lettuce | Yes | 350 | | *Chagvardieff et al., 1994* |
| *Lactuca sativa* 'MV' | lettuce | Yes | 308 | | *Baslam et al., 2012* |
| *Lactuca sativa* 'Waldmann's Green' | lettuce | Yes | 600 | | *McKeehen et al., 1996* |
| *Lycopersicon esculentum* 'Astra' | tomato | Yes | 600 | | *Khan et al., 2013* |
| *Lycopersicon esculentum* 'Eureka' | tomato | Yes | 600 | | *Khan et al., 2013* |
| *Lycopersicon esculentum* 'Mill.' | tomato | Yes | 360 | | *Li et al., 2007* |
| *Lycopersicon esculentum* 'Zheza 809' | tomato | Yes | 450 | | *Jin et al., 2009* |
| *Mangifera indica* 'Kensington' | mango tree | Yes | 350 | | *Schaffer and Whiley, 1997* |
| *Mangifera indica* 'Tommy Atkins' | mango tree | Yes | 350 | | *Schaffer and Whiley, 1997* |
| *Medicago sativa* | alfalfa | Yes | 615 | | *Tian et al., 2014* |
| *Medicago sativa* 'Victor' | alfalfa | Yes | 100 | UK | *Al-Rawahy et al., 2013* |
| *Oryza sativa* | rice | Yes | 200 | China | *Pang et al., 2005* |
| *Oryza sativa* 'Akitakomachi' | rice | Yes | 205–260 | Japan | *Lieffering et al., 2004* |
| *Oryza sativa* 'Akitakomachi' | rice | Yes | 250 | Japan | *Yamakawa et al., 2004* |
| *Oryza sativa* 'BRRIdhan 39' | rice | Yes | 210 | Bangladesh | *Razzaque et al., 2009* |
| *Oryza sativa* 'Gui Nnong Zhan' | rice | Yes | 500 | | *Li et al., 2010* |
| *Oryza sativa* 'IR 72' | rice | Yes | 296 | Philippines | *Ziska et al., 1997* |
| *Oryza sativa* 'Japonica' | rice | Yes | 200 | China | *Jia et al., 2007* |
| *Oryza sativa* 'Jarrah' | rice | Yes | 350 | | *Seneweera and Conroy, 1997* |
| *Oryza sativa* 'Khaskani' | rice | Yes | 210 | Bangladesh | *Razzaque et al., 2009* |
| *Oryza sativa* 'Rong You 398' | rice | Yes | 500 | | *Li et al., 2010* |
| *Oryza sativa* 'Shakkorkhora' | rice | Yes | 210 | Bangladesh | *Razzaque et al., 2009* |
| *Oryza sativa* 'Shan You 428' | rice | Yes | 500 | | *Li et al., 2010* |
| *Oryza sativa* 'Tian You 390' | rice | Yes | 500 | | *Li et al., 2010* |
| *Oryza sativa* 'Wu Xiang jing' | rice | Yes | 200 | China | *Guo et al., 2011* |
| *Oryza sativa* 'Wuxiangjing 14' | rice | Yes | 200 | China | *Ma et al., 2007* |
| *Oryza sativa* 'Wuxiangjing 14' | rice | Yes | 200 | China | *Yang et al., 2007* |
| *Oryza sativa* 'Yin Jing Ruan Zhan' | rice | Yes | 500 | | *Li et al., 2010* |
| *Oryza sativa* 'Yue Za 889' | rice | Yes | 500 | | *Li et al., 2010* |
| *Phaseolus vulgaris* 'Contender' | bean | Yes | 340 | | *Mjwara et al., 1996* |
| *Phaseolus vulgaris* 'Seafarer' | bean | Yes | 870 | | *Porter and Grodzinski, 1984* |
| *Raphanus sativus* 'Mino' | radish | Yes | 600 | | *Azam et al., 2013* |
| *Raphanus sativus* 'Cherry Belle' | radish | Yes | 380 | | *Barnes and Pfirrmann, 1992* |
| *Raphanus sativus* 'Giant White Globe' | radish | Yes | 600 | | *McKeehen et al., 1996* |
| *Rumex patientia x R. Tianschanicus* 'Rumex K-1' | buckwheat | Yes | 615 | | *Tian et al., 2014* |

Table 3. Continued on next page

*Table 3. Continued*

| Species | Common name | Crop | +CO2 | Country | Reference |
|---|---|---|---|---|---|
| *Secale cereale* 'Wintergrazer-70' | rye | Yes | 615 | | *Tian et al., 2014* |
| *Solanum lycopersicum* '76R MYC+' | tomato | Yes | 590 | | *Cavagnaro et al., 2007* |
| *Solanum lycopersicum* 'rmc' | tomato | Yes | 590 | | *Cavagnaro et al., 2007* |
| *Solanum tuberosum* | potato | Yes | 500 | | *Cao and Tibbitts, 1997* |
| *Solanum tuberosum* 'Bintje' | potato | Yes | 170 | Germany | *Högy and Fangmeier, 2009* |
| *Solanum tuberosum* 'Bintje' | potato | Yes | 278-281 | Sweden | *Piikki et al., 2007* |
| *Solanum tuberosum* 'Bintje' | potato | Yes | 305-320 | Europe | *Fangmeier et al., 2002* |
| *Solanum tuberosum* 'Dark Red Norland' | potato | Yes | 345 | USA | *Heagle et al., 2003* |
| *Solanum tuberosum* 'Superior' | potato | Yes | 345 | USA | *Heagle et al., 2003* |
| *Sorghum bicolor* | sorghum | Yes | 360 | USA | *Prior et al., 2008* |
| *Spinacia oleracea* | spinach | Yes | 250 | India | *Jain et al., 2007* |
| *Trigonella foenum-graecum* | fenugreek | Yes | 250 | India | *Jain et al., 2007* |
| *Triticum aestivum* | wheat | Yes | 175 | Germany | *Erbs et al., 2010* |
| *Triticum aestivum* 'Ningmai 9' | wheat | Yes | 200 | China | *Ma et al., 2007* |
| *Triticum aestivum* 'Triso' | wheat | Yes | 150 | Germany | *Högy et al., 2009* |
| *Triticum aestivum* 'Triso' | wheat | Yes | 150 | Germany | *Högy et al., 2013* |
| *Triticum aestivum* 'Alcazar' | wheat | Yes | 350 | | *de la Puente et al., 2000* |
| *Triticum aestivum* 'Batis' | wheat | Yes | 170 | Germany | *Wroblewitz et al., 2013* |
| *Triticum aestivum* 'Dragon' | wheat | Yes | 305-320 | Sweden | *Pleijel and Danielsson, 2009* |
| *Triticum aestivum* 'HD-2285' | wheat | Yes | 250 | India | *Pal et al., 2003* |
| *Triticum aestivum* 'Janz' | wheat | Yes | 166 | Australia | *Fernando et al., 2014* |
| *Triticum aestivum* 'Jinnong 4' | wheat | Yes | 615 | | *Tian et al., 2014* |
| *Triticum aestivum* 'Minaret' | wheat | Yes | 278 | Germany | *Fangmeier et al., 1997* |
| *Triticum aestivum* 'Minaret' | wheat | Yes | 300 | Europe | *Fangmeier et al., 1999* |
| *Triticum aestivum* 'Rinconada' | wheat | Yes | 350 | | *de la Puente et al., 2000* |
| *Triticum aestivum* 'Star' | wheat | Yes | 334 | Germany | *Manderscheid et al., 1995* |
| *Triticum aestivum* 'Turbo' | wheat | Yes | 334 | Germany | *Manderscheid et al., 1995* |
| *Triticum aestivum* 'Turbo' | wheat | Yes | 350 | | *Wu et al., 2004* |
| *Triticum aestivum* 'Veery 10' | wheat | Yes | 410 | | *Carlisle et al., 2012* |
| *Triticum aestivum* 'Yangmai' | wheat | Yes | 200 | China | *Guo et al., 2011* |
| *Triticum aestivum* 'Yitpi' | wheat | Yes | 166 | Australia | *Fernando et al., 2012a* |
| *Triticum aestivum* 'Yitpi' | wheat | Yes | 166 | Australia | *Fernando et al., 2012b* |
| *Triticum aestivum* 'Yitpi' | wheat | Yes | 166 | Australia | *Fernando et al., 2012c* |
| *Triticum aestivum* 'Yitpi' | wheat | Yes | 166 | Australia | *Fernando et al., 2014* |

The table provides species name, common name, the type of experimental set up, the level of $CO_2$ enrichment, and indicates whether the species is a crop. Countries are listed only for FACE and OTC type experiments with 'Europe' accounting for combined data from Belgium, Denmark, Finland, Germany, Sweden, and the UK.

or more separate datasets for the same species or cultivar, the data were averaged prior to the inclusion into the foliar dataset. For the edible tissue dataset, the study inclusion rules were the same as for the foliar dataset with the following exception: due to relative scarcity of data for edible tissues, the data with co-factors were included in the dataset (e.g., observations from combined $eCO_2$ and ozone experiments were included). The 'Additional info' column in the dataset specifies exactly what datasets were extracted from each study with multiple datasets.

The above publication-inclusion and data-inclusion rules allow treating each study as independent in the dataset. At no instance, potentially co-dependent observations (e.g., multiple observations of the same plant throughout a growing season or observations of various parts of the same plant) were included in either the foliar or the edible dataset as separate studies. I used GraphClick v.3.0 and PixelStick v.2.5 to digitize data presented in a graphical form, for example bar charts.

The foliar dataset covers 4733 observations of 25 chemical elements in 110 species and cultivars. The edible tissues dataset covers 3028 observations of 23 elements in 41 species and cultivars. The FACE studies cover 2264 observations of 24 elements in 25 species and cultivars. The two datasets reflect data on 125 $C_3$ and 5 $C_4$ species/cultivars.

## Effect size measure

While the amount of statistical details provided in each study varies considerably, the following data were extractable from each study: (1) the relative change (or lack thereof) in the mean concentration between the low and the high $CO_2$ treatments: $(E-A)/A$, where $A$ and $E$ are the mean concentrations of an element at the low and the high $CO_2$ treatments respectively, (2) the sample size or the number of replicates ($n$).

Since a decrease in the concentration of a mineral is limited to 100%, but an increase in its concentration is theoretically unlimited, a standard technique was applied to reduce biases towards increases. Specifically, the natural log of the response ratio, that is $ln(E/A)$, was used as the effect size metric (e.g., *Hedges et al., 1999*; *Jablonski et al., 2002*; *Taub et al., 2008*). The response ratio, $r = E/A$, was calculated from the relative change as follows: $r = 1+(E-A)/A$. After performing statistical analyses, I converted all the results back from the log form to report them as ordinary percent changes.

## Making results replicable

Published meta-analytic and biostatistical results need to be replicable and reproducible, and the process of replication needs to be made as easy as possible and clearly traceable to the original sources (*Peng, 2009*). In this regard, I have made the following efforts to ease the replication (from the original sources) of each and every result presented here:

1. While copyright restrictions do not permit posting the original published data sources online, I will share, upon request, all the data sources in PDF form, where all the pertinent data are clearly marked for easy identification, thus removing any potential ambiguity about what data were extracted from each study.
2. The entire dataset for the foliar and the edible tissues is available at Dryad digital depository, www.datadryad.org, under 10.5061/dryad.6356f. The dataset is available as an Excel file (formatted for easy viewing) and as a 'CSV' file; the latter is made-ready (tidy) for analysis with open-source (*R Core Team, 2014*) and commercial statistical packages (e.g., SPSS).
3. An executable R code to generate individual results is available with the dataset at the above-mentioned depository and at GitHub: https://github.com/loladze/co2. Assistance for replicating any result and figure presented in this study will be provided to any interested party.

## Statistical analysis

I performed all the analyses using R (*R Core Team, 2014*), SPSS v. 21 (IBM, Armonk, NY, USA) and G*Power 3 (*Faul et al., 2007*). Meta-analytic studies often weight effect sizes by the reciprocal of their variance, which tends to give a greater weight to studies with greater precision. However, many $eCO_2$ studies do not report measures of variation in the data (standard error, standard deviation, or variance). In lieu of the measures of variance, studies can be weighted by the number of replicates ($n$) or, alternatively, each study can be assigned equal weight, that is, unweighted method (*Jablonski et al., 2002*). I used both methods (weighted and unweighted) to calculate the means of effect sizes with 95% CIs and compared the results of both methods. Nearly in all instances, the difference between the weighted and the unweighted means was small and lesser than the standard error of the unweighted mean. For example, across all the FACE studies, the overall mineral change was −6.1% (−7.8 to −4.4) when unweighted cf. the −6.5% (−8.0 to −5.1) when weighted. For the reason of close similarity between weighted and unweighted approaches, I used the simpler out of the two methods, that is the unweighted one, when reporting the results.

Since the distribution of effect sizes is not necessarily normal, I applied both parametric ($t$ test) and non-parametric (bootstrapping with 10,000 replacements) tests for calculating the 95% CI for the

mean effect size and the statistical power. The latter was calculated for: (1) an absolute effect size of 5%, and (2) the probability of Type I error, $\alpha = 0.05$. If the variance of a small sample << the true population variance, then this leads to substantial overestimations of Cohen's $d$ and the statistical power. To be conservative when estimating power for small samples ($m <20$), I used the *larger* of the sample standard deviation or 0.21, which is the standard deviation for the entire mineral dataset.

The results from the parametric and non-parametric tests were very close. For example, for Zn in edible tissues (sample size = 65), $t$ test yields ($-11.4\%$, $-14.0$ to $-8.7$) and 0.91 power cf. ($-11.4\%$, $-13.9$ to $-8.7$) and 0.92 power for the bootstrapping procedure. A close similarity between the results of $t$ test and non-parametric test is expected when sample size ($m$, the number of independent observations for each mineral) is >30, which often was the case in this study. For reporting purposes, I used the 95% CI and the power generated by the non-parametric method, that is, the bootstrapping procedure.

## Testing for publication bias

To test for publication bias or 'the file drawer effect' in the dataset, I plotted effect sizes against corresponding sample sizes/replicates, $n$, to provide a simple visual evaluation of the distribution of effect sizes (*Figure 9*). The resulting cloud of points is funnel-shaped, narrowing toward larger sample sizes, and overall is symmetrical along the mean effect size. This indicates the absence of any significant publication bias (*Egger et al., 1997*).

## Fragmenting the dataset into categories

Meta-analytic $CO_2$ studies often partition their datasets into various categories (e.g., plant group, plant tissue, fertilization, or water regime) to estimate effect sizes for each category. Such data fragmentation, however, is warranted only if the statistical power of the resulting test for each category is adequate. Otherwise, low power can lead to non-significant outcomes and Type II errors. As tempting as it can be to partition the current dataset into many categories and cases (e.g., Zn in fruits, Fe in tuber, Cu in annuals, multiple $CO_2$ levels), only by fragmenting the data into sufficiently large categories an adequate statistical power can be retained. Such categories include: foliar tissues, edible tissues, woody plants (trees and shrubs), herbaceous plants, FACE studies, non-FACE studies, crops, wild plants (all non-crops, including ornamental plants), $C_3$ plants, $C_4$ plants, rice, wheat, barley, and potato. Furthermore, I fragmented the data for $C_3$ plants, the foliar and the edible tissues, the non-FACE and the FACE studies into individual chemical elements and into individual common plant names (e.g., all rice cultivars grouped under 'rice'). For the regional analysis, I used only OTC and FACE studies because they reflect local environment much more accurately than studies using complete-enclosures (e.g., closed chamber, glasshouse). If an OTC or FACE study did not report precise geographic coordinates, then the latitude and longitude of a nearby research facility or city was used (all coordinates in the dataset are in decimal units). *Figures 1–7* include results with the statistical power >0.40 for each element, country, region, plant tissue or category. Generally, power >0.80 is considered acceptable (*Cohen, 1988*). Unfortunately, such a level was achievable only for elements for which the data are most abundant and for the ionomes of some plant groups and species. Note that the power was calculated for a 5% effect size, while the true effect size is likely to be larger (~8%); therefore, the true power is likely to be higher than the calculated power for most results. All the results, irrespective of the statistical power, can be found in *Figure 1–source data 1*. Furthermore, *Figure 1* shows the mean effect sizes (with their 95% CI) plotted against their respective statistical powers for all the minerals and all the plant groups/tissues.

## Formula for calculating percentage changes in TNC:protein and TNC:minerals

If the concentration of substance X in a plant increases by $x$% and concomitantly the concentration of substance Y decreases by $y$% in the plant, then the X-to-Y ratio of the plant (X:Y) increases by:

$$\frac{x + y}{100 - y} \cdot 100\% \tag{1}$$

## Proof

Let us denote the initial concentrations of substances $X$ and $Y$ in a plant as $x_A$ and $y_A$, respectively. Suppose the $X$ and $Y$ contents in the plant changed by $x$% and $-y$%, respectively. Then the new $X$ content in the plant, $x_E$, is

$$x_E = x_A \cdot (100 + x)\,\%,$$

and the new $Y$ content in the plant, $y_E$, is

$$y_E = y_A \cdot (100 - y)\,\%.$$

The original $X : Y = x_A/y_A$, while the new $X : Y = x_E/y_E$. Since the percentage change in the $X{:}Y$ equals to:

$$\frac{new - original}{original} \cdot 100\% = \left(\frac{new}{original} - 1\right) \cdot 100\%,$$

substituting $x_A/y_A$ and $x_E/y_E$ for the original and the new, respectively, yields:

$$\frac{x_E/y_E}{x_A/y_A} - 1 = \frac{x_E \cdot y_E}{x_A \cdot y_A} - 1 = \frac{(x_A\,(100 + x)\,\%) \cdot y_A}{x_A\,(y_A\,(100 - y)\,\%)} - 1 = \frac{100 + x}{100 - y} - 1 = \frac{x + y}{100 - y}.$$

An advantage of *Equation 1* is that it holds true irrespective of whether the decrease in $Y$ is driven by some reason applicable only to $Y$ or by the increase in $X$, that is the dilution by $X$.

## Acknowledgements

The author thanks George Kordzakhia, Nik Loladze and Marina Van for discussions, David Salt and four anonymous referees for comments, and Dmitri Logvinenko for providing access to library resources. The author acknowledges NSF rejections to support this research (proposals Nos. 0548181, 0644300, 0746795).

## Additional information

### Funding

| Funder | Author |
| --- | --- |

The author declares that there was no external funding for this work.

### Author contributions

IL, Conception and design, Acquisition of data, Analysis and interpretation of data, Drafting or revising the article

## Additional files

### Major dataset

The following dataset was generated:

| Author(s) | Year | Dataset title | Dataset ID and/or URL | Database, license, and accessibility information |
| --- | --- | --- | --- | --- |
| Loladze I | 2014 | $CO_2$ Dataset (CSV format); $CO_2$ Dataset (XLSX format); R Code for the $CO_2$ dataset | http://dx.doi.org/10.5061/dryad.6356f | Available at Dryad Digital Repository under a CC0 Public Domain Dedication. |

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
