## [Decision Letter]

eLife posts the editorial decision letter and author response on a selection of the published articles (subject to the approval of the authors). An edited version of the letter sent to the author after peer review is shown, indicating the substantive concerns or comments; minor concerns are not usually shown. Reviewers have the opportunity to discuss the decision before the letter is sent (see review process). Similarly, the author response typically shows only responses to the major concerns raised by the reviewers.

[Editors’ note: a previous version of this study was rejected after peer review, but the author submitted for reconsideration. The two decision letters after peer review are shown below.]

Thank you for choosing to send your work entitled “Hidden shift: elevated CO_2_ alters the plant ionome depleting minerals at the base of human and herbivore nutrition” for consideration at *eLife*. Your full submission has been evaluated by a Senior editor and 3 peer reviewers, and the decision was reached after discussions between the reviewers. We regret to inform you that your work will not be considered further for publication at this time.

The following peer reviewers have agreed to reveal their identity: David Salt and Lisa Ainsworth.

This is not an easy call, as all reviewers agreed that it was a solid analysis that builds significantly on the previously published work in Trends in Ecology and Evolution in 2002 and that the power analysis was a particularly noteworthy advance. The reason for decision to reject lay in the concerns about the “scaleability” the results from the FACE trails to human nutrition. The conclusions were based on analogies to human obesity studies and were simply too strongly drawn to be supported by the data. It also wasn't clear to the reviewers that the FACE results could extrapolate to tropical agricultural systems given that tropical agricultural productivity is limited by other factors (water, nitrogen, pests etc).

*Reviewer*
*#1:*

This manuscript details a meta analysis based on published data on the concentration of elements (aka ionome) in various plant tissues and species from studies in which atmospheric CO_2_ has been varied. Analysis of the data in an appropriate statistical framework revealed significant chances in the plant ionome after growth of plants in atmospheres with elevated CO_2_ (in both laboratory and field-based experiments). Many of these changes were not observed as significant changes in the original studies due to low sample sizes. Further analyses by integration of published data on carbohydrate content of plant tissues reveals that these changes in the plant ionome are likely due to dilution by the enhanced accumulation of carbohydrates observed when plants are grown in elevated CO_2_. An interesting discussion is then presented on the potential significance of this dilution of essential mineral nutrients in our global food supply.

I enjoyed reading this manuscript and liked the discursive style (something that is now quite rare in the scientific literature). However, I felt the manuscript was too long and both the Introduction and Discussion could be significantly shortened after careful editing without significant loss of readability or information content. For example, the long discussion on sample size being important to detect significant differences between treatments when the effect is expected to be small could be significantly reduced.

*Reviewer*
*#2:*

This is an interesting review of the effects of rising CO_2_ on the mineral content of plants. This author previously published a study on this topic, alerting the community to the detrimental impact that rising CO_2_ concentrations were having on mineral content of plants and edible parts of plants. In the current manuscript, a much larger data set is compiled and statistically analyzed to report that elevated CO_2_ significantly decreases mineral content in leaves and other edible parts of plants. Much thought and discussion is given to the power of the meta-dataset and I think that this is an important aspect of the paper. Finally, a thought experiment is done to discuss the potential impact of the increase in C and decrease in mineral nutrients on human health.

I think that the biggest question from this analysis is the impact on human health. In regions of the world where people are most dependent on bioavailable calories and nutrients from plants, few elevated CO_2_ experiments have been done. For example, there are no published data from FACE experiments in the tropics. In tropical regions, drought, extreme temperatures and/or very poor nutrient supply likely limit agricultural production and in these areas elevated CO_2_ may have substantially less impact on plant growth or plant quality. Therefore, it is very uncertain what effect elevated CO_2_ will have on human nutrition there, and I think this needs to be acknowledged as a gap in the data and in the potential inferences made in this paper.

*Reviewer*
*#3:*

I very much enjoyed reading this paper. It takes a clever approach to the highly significant issue of how climate change might impact on the human food chain via its influence of plant composition, and in so doing does an excellent job of discussing the results in a broad integrative context. Usually a critical reviewer, I could find little to complain about here: the story is important, convincing, and nicely told.

[Editors’ note: what now follows is the decision letter after the authors submitted for further consideration.]

Thank you for sending your work entitled “Hidden shift: elevated CO_2_ alters the plant ionome and depletes minerals at the base of human nutrition” for further consideration at *eLife*. Your article has been favorably evaluated by Ian Baldwin and 3 new peer reviewers.

The Reviewing editor and the reviewers discussed their comments before we reached this decision, and the Reviewing editor has assembled the following comments to help you prepare a revised submission.

This manuscript presents a unique collection of data on CO_2_-induced changes of the plant ionome, which clearly show that the majority of plants investigated so far showed a remarkably similar tendency in their response to CO_2_ (albeit with variation). This is clearly an under-appreciated component of the undeniable rise in global CO_2_ levels, which deserves more attention. The reviewers also recognized that the core of the argument that relates to the impact of changes in nutrient content of the edible portion of food crops on human health has simply not been settled, and were (again) split as to whether this problem was sufficient to reject the manuscript for *eLife*. After discussion, a consensus agreement was reached that the manuscript could be accepted if it was substantially revised so that it was clear that impact of changes in nutrient content of the edible portion of food crops on human health has not been settled. We hope that in revising the manuscript, this uncertainty is explicitly addressed and that you could highlight the need for more research to address this very important but festering issue. In addition, it was felt that the Introduction should be shortened, downplaying the thought experiment, and significantly tempering the conclusions drawn in the Discussion.

---

## [Author Response]

[Editors’ note: the author responses to the first round of peer review follow.]

Reviewer 2 claims that there is “no published data from FACE experiments in the tropics.” Her opinion is that “elevated CO_2_ may have substantially less impact” on plant quality in the tropics and, “therefore, it is very uncertain what effect elevated CO2 will have on human nutrition there.”

This argument is flawed. There *are* published FACE, open-top chamber and greenhouse experiments carried out between the 35**°** N & S latitudes – the tropical and subtropical regions, where large parts of malnourished population reside (e.g., [93]; [68], [110], [109]; [146]; Khan *et al.* 2012; Azam *et al.* 2012), and they do show declines in the plant mineral content. Prompted by the *eLife* review, I made the regional analysis of all the CO_2_ studies carried out between the 35**°** N & S parallels: it shows that the plant mineral content declines by 5% in the region. Furthermore, many countries in the tropics rely on imports of wheat, maize and soybeans, most of which are grown north of the N 35**°**parallel, where FACE and other experiments also reveal declines in the crop mineral content.

No reviewer found any logical flaws in my human-nutrition thought experiment, which relies on the rigor of mass balance laws. However, I understand that such “experiments” are not conventional even if their conclusions are valid. For this reason, I can tone down and shorten the health and obesity discussion. The revised paper will focus on firmly establishing a novel and important aspect of global change – the shift in the plant ionome induced by the rising CO_2_.

I emphasize that this is a novel result because the last definitive word on the issue was [29] meta-analysis claiming the absence of any prevailing effect of elevated CO_2_ on the plant minerals and, specifically, the lack of response of grain minerals to high-CO_2_ – claims that are opposite to my results.

The power is in your hands to give my revised and stronger paper further consideration at e*Life* and to advance the progress on this important issue.

[Editors’ note: the author responses to the re-review follow.]

*After discussion, a consensus agreement was reached that the manuscript could be accepted if it was substantially revised so that it was clear that impact of changes in nutrient content of the edible portion of food crops on human health has not been settled. We hope that in revising the manuscript, this uncertainty is explicitly addressed and that you could highlight the need for more research to address this very important but festering issue*.

I revised the manuscript accordingly. Specifically:

I have added the following statement to the Discussion: “I emphasize that the impact of CO_2_-induced shifts in the quality of crops on human health is far from settled. The purpose of what follows is not to make definitive claims but to stimulate research into this important but festering issue.”

I have added a new subsection to the Discussion, titled “Data Scarcity,” noting that for many crops, the pertinent data are limited or non-existent.

The wording about the effects of the mineral decline on human nutrition was toned down from “will” to “can”/“might/potential”. (Furthermore, the abstract was changed to stress that the effects on human health are discussed (and, thus, are not parts of results.)

*In addition, it was felt that the Introduction should be shortened, downplaying the thought experiment..*.

I deleted the passage referring to my 2002 ‘thought experiment’ from the Introduction and do not mention it anywhere else in the manuscript.

I deleted the reference to Loladze & Elser (2011) together with the sentence stating that the cellular stoichiometric homeostasis is sensitive to the environment.

I have shortened and simplified the passage about the dichotomy between CO_2_ effects on N and minerals.

I have shortened and improved readability of the list of questions at the end of the Introduction. The new Introduction is shorter by ∼200 words.

*...and significantly tempering the conclusions drawn in the Discussion*.

Aside from changes indicated above, I have made the following changes to the Discussion:

1) Deleted Figure 9 showing the graphical output of [49] dynamic model of weight gains in a female and a male.

2) Tempered and shortened conclusions about the impact on human health. Now the statements are reduced to: “The above ‘experiment’ suggests that a systemic and sustained 5% mineral depletion in plants can be nutritionally significant. While the rise in the atmospheric CO_2_ concentration is expected to be nearly uniform around the globe, its impact on crop quality might unequally affect the human population: from no detrimental effects for the well-nourished people to potential weight gain for the calorie-sufficient but mineral-undernourished.”

In addition, to the above changes I revamped the Results by separating them into clear subsections. Furthermore, the readability throughout the paper was improved.